 **eLife**

# Autophagy linked FYVE (Alfy/WDFY3) is required for establishing neuronal connectivity in the mammalian brain

Joanna M Dragich[1], Takaaki Kuwajima[2,3], Megumi Hirose-Ikeda[1], Michael S Yoon[1], Evelien Eenjes[1], Joan R Bosco[1], Leora M Fox[1,4], Alf H Lystad[5], Tinmarla F Oo[1], Olga Yarygina[1], Tomohiro Mita[6], Satoshi Waguri[7], Yoshinobu Ichimura[6], Masaaki Komatsu[6], Anne Simonsen[5], Robert E Burke[1,2,3], Carol A Mason[2,3,8,9], Ai Yamamoto[1,2,3]*

[1]Department of Neurology, College of Physicians and Surgeons, Columbia University, New York, United States; [2]Department of Pathology, College of Physicians and Surgeons, Columbia University, New York, United States; [3]Department of Cell Biology, College of Physicians and Surgeons, Columbia University, New York, United States; [4]Doctoral Program in Neurobiology and Behavior, Columbia University, New York, United States; [5]Institute of Basic Medical Sciences, University of Oslo, Oslo, Norway; [6]The Protein Metabolism Project, Tokyo Metropolitan Institute of Medical Science, Tokyo, Japan; [7]Department of Anatomy and Histology, Fukushima Medical University School of Medicine, Fukushima, Japan; [8]Department of Neuroscience, College of Physicians and Surgeons, Columbia University, New York, United States; [9]Department of Ophthalmology, College of Physicians and Surgeons, Columbia University, New York, United States

*For correspondence: ay46@cumc.columbia.edu

Competing interests: The authors declare that no competing interests exist.

**Abstract** The regulation of protein degradation is essential for maintaining the appropriate environment to coordinate complex cell signaling events and to promote cellular remodeling. The Autophagy linked FYVE protein (Alfy), previously identified as a molecular scaffold between the ubiquitinated cargo and the autophagic machinery, is highly expressed in the developing central nervous system, indicating that this pathway may have yet unexplored roles in neurodevelopment. To examine this possibility, we used mouse genetics to eliminate Alfy expression. We report that this evolutionarily conserved protein is required for the formation of axonal tracts throughout the brain and spinal cord, including the formation of the major forebrain commissures. Consistent with a phenotype reflecting a failure in axon guidance, the loss of Alfy in mice disrupts localization of glial guidepost cells, and attenuates axon outgrowth in response to Netrin-1. These findings further support the growing indication that macroautophagy plays a key role in the developing CNS.

## Introduction

The Autophagy linked FYVE domain protein (Alfy) [gene name, WD40 repeat and FYVE domain protein 3 (*Wdfy3*)] is a member of the Beige and Chediak-Higashi (BEACH) domain containing proteins, a family of proteins implicated in vesicle trafficking and membrane dynamics (*Isakson et al., 2013*; *Simonsen et al., 2004*). As its name implies, Alfy has been implicated in the degradative pathway macroautophagy, by acting as a molecular scaffold between select cargo and core members of the mammalian autophagic machinery such as Atg5, p62 and Atg8 homologs (*Clausen et al., 2010*; *Filimonenko et al., 2010*; *Lystad et al., 2014*; *Simonsen et al., 2004*). In addition, its functional

**eLife digest** Unlike many other cells in the body, neurons typically survive throughout the life of a mammal. This long life suggests that they may be more vulnerable to damage from cellular debris. Previous research has found that a protein called Alfy, which is abundant in the brain, is involved in cleaning up debris, such as those involved in neurodegenerative diseases, by a pathway known as autophagy. Alfy guides the formation of spherical compartments called autophagosomes, which deliver the debris to another compartment known as the lysosome to permit degradation.

In developing embryos, neurons need to migrate to the right location within the central nervous system and extend projections called axons to communicate with other cells. However, it was not clear whether this process requires cell materials to be selectively sent to lysosomes, and whether this involves the Alfy protein.

Dragich et al. addressed this question by studying mouse embryos that lack Alfy. The brains of these mice developed abnormally and were missing the corpus callosum (the dense band of fibers that normally connects the two halves of the brain). Without Alfy, the growing axons could not navigate their way to the right places to connect with other neurons. Furthermore, some neurons migrated to the wrong places in the developing brain, which resulted in the abnormal formation of cell-clusters.

The findings of Dragich et al. suggest that autophagy also plays an important role in normal brain development. Future studies are now needed to work out exactly how Alfy controls neuron migration and the growth of axons. The human gene *WDFY3* is nearly identical to the gene that encodes the Alfy protein, and has been implicated in neurodevelopmental disorders such as autism and microencephaly. Studying Alfy therefore may help us to understand human conditions that affect the developing or aging brain.

FYVE zinc finger domain at the COOH-terminus permits partial co-localization to phosphatidylinositol-3-monophosphate (PtdIns3P), especially at autophagosome membranes (*Simonsen et al., 2004*).

*Wdfy3* is evolutionarily conserved and the most extensively studied homolog is *Blue Cheese (bchs)* in *D. melanogaster* (*Finley et al., 2003*). In the developing and adult fly central nervous system (CNS), Bchs is abundantly expressed, with preferential accumulation in axon terminals and at the growth cone (*Finley et al., 2003*; *Khodosh et al., 2006*). Adult *bchs* null flies have a shortened life span and show signs of adult onset neurodegeneration, including the accumulation of ubiquitinated aggregates (*Filimonenko et al., 2010*; *Finley et al., 2003*; *Khodosh et al., 2006*). Loss-of-function (LoF) mutations in *bchs* disrupt the axonal transport of endolysosomal vesicles (*Lim and Kraut, 2009*), however no defects in axon guidance have been reported in *bchs* null larva (*Khodosh et al., 2006*). Recently it has been reported that in vertebrates, genetically diminished levels of Alfy disrupts neurogenesis leading to altered forebrain morphology (*Orosco et al., 2014*). Furthermore, genetic screening has revealed a possible role for the human homolog *WDFY3* as a genetic risk factor for intellectual and developmental disabilities (IDD), microcephaly and neuropsychiatric disorders (*Bonnet et al., 2010*; *Iossifov et al., 2012*; *Kadir et al., 2016*). These findings raise the possibility that Alfy could have an important function in mammalian CNS development.

Here, we present two new mouse models that eliminate Alfy expression and identify an essential role for Alfy during murine development. Constitutive elimination of Alfy leads to perinatal lethality, in conjunction with developmental brain wiring defects throughout the CNS, involving forebrain commissures, internal capsule, optic chiasm, spinal cord and longitudinal tracts such as the medial forebrain bundle. In the ventral midbrain, dopaminergic cell populations retain an immature morphology and their axons aberrantly project into the hypothalamic region, forming an ectopic commissure near the optic chiasm. Consistent with a failure of axon guidance mechanisms, localization of glial guidepost cells for callosal axons were disrupted, and sensitivity of Alfy knockout axons to the trophic effect of Netrin-1 was significantly diminished. Moreover, Alfy is enriched in membrane fractions, suggesting that it may play a key role in membrane trafficking events to establish neural connectivity in the mammalian brain.

## Results

### Alfy is highly expressed in the CNS

To characterize the role of Alfy in mouse, we initially determined when and where Alfy/Wdfy3 is expressed. Multiplex, semi-quantitative RT-PCR revealed that *Wdfy3* mRNA could be detected as early as embryonic day (E) 11 in CNS tissue, and remains detectable throughout gestation (*Figure 1A*). Similar analysis in adult tissue revealed that the *Wdfy3* transcript is ubiquitously expressed, and that the highest concentration of Alfy was observed in the brain (*Figure 1—figure supplement 1*), confirming previous results (*Simonsen et al., 2004*). *Wdfy3* transcript is detected throughout the both the perinatal and adult brain, as determined by *in situ* hybridization (ISH) (*Figure 1B* and not shown). Immunoblotting revealed that expression of the protein was present uniformly throughout the brain (*Figure 1C*). Using both primary neuronal and purified astroglial cultures, endogenous Alfy expression was detected in both cell types (*Figure 1—figure supplement 2*), supporting recent transcriptome analysis of the mouse cortex (*Zhang et al., 2014*). Therefore, we conclude that Alfy is a CNS-enriched protein that is present in various neuronal and non-neuronal cell types in the developing and adult brain.

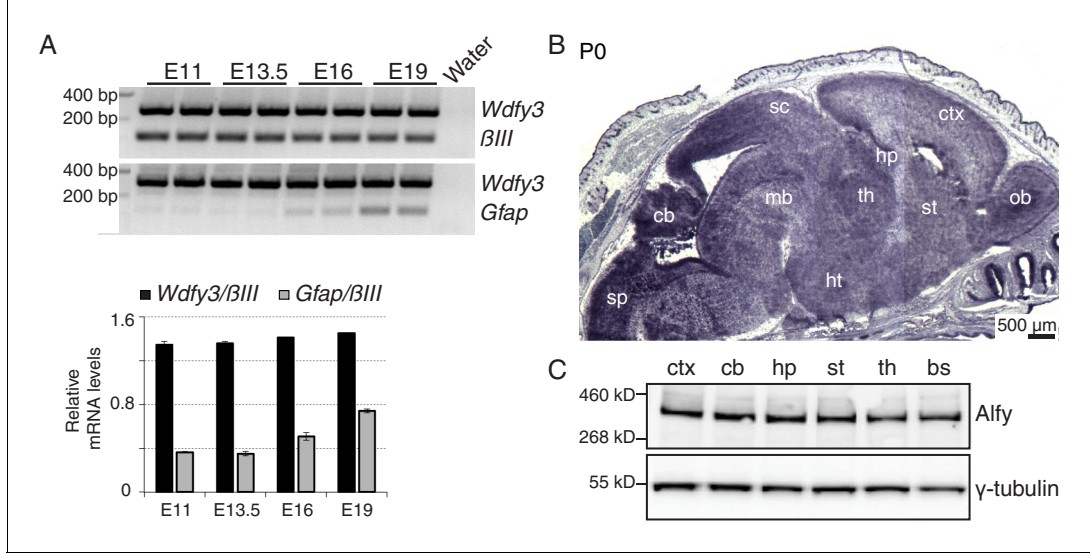

**Figure 1.** Alfy is highly expressed throughout the developing and adult mouse CNS. (**A**) (Top) RT-PCR demonstrates Alfy/Wdfy3 can be detected as early as embryonic day 11 (E11) and remains abundant throughout gestation (E19). (**A**) Multiplex PCR for the 5' region of the gene encoding Alfy, *Wdfy3* with *Class III β-tubulin* (*βIII*, top) and *Glial fibrillary acid protein* (*Gfap*, bottom). (Btm) Quantification of transcript levels relative to *βIII* (black) and *Gfap* (gray), n = 4, bars represent mean ± SEM. (**B**) *In situ* hybridization of sections from P0 mice reveals strong expression of *Wdfy3* throughout the brain. Low magnification images stitched together to reveal a complete sagittal brain section representative of *Wdfy3* mRNA distribution in a wildtype mouse. An abundance of mRNA is observed throughout the newborn CNS (n = 4), whereas no mRNA staining above background is observed in Alfy KO mice (data not shown). The ISH reflects how Alfy is most highly expressed in brain (*Figure 1—figure supplement 1*) and this expression is both neuronal and glial (*Figure 1—figure supplement 2*). Abbreviations: cerebellum (cb), cerebral cortex (ctx), hippocampus (hp), hypothalamus (ht), midbrain (mb), olfactory bulb (ob), striatum (st), superior colliculus (sc) and thalamus (th). (**C**) Immunblotting reveals that Alfy expression is maintained throughout the adult brain. Lysates generated from different regions of the adult brain, including the brain stem (bs) were probed with an Alfy antibody raised against its N-terminus. γ-tubulin is shown as a loading control (n = 4).

The following figure supplements are available for figure 1:

**Figure supplement 1.** Alfy/Wdfy3 expression is highest in the brain.

**Figure supplement 2.** Alfy is expressed in neurons and astroglia.

## Loss of Alfy expression results in perinatal lethality

To investigate the consequence of the genetic deletion of *Wdfy3* in mice, we generated and characterized two different Alfy deficient mouse lines: One using gene trap (GT)-mediated disruption and a second using a conditional strategy (*Figure 2*, *Figure 2—figure supplement 1*). Whereas several GT lines disrupting the *Wdfy3* locus were found, one that contained a GT insertion within the first intron was predicted to completely abolish production of the full length transcript. To confirm, mice heterozygous for this mutation (Alfy GT Het) were interbred. Homozygous mice (Alfy GT) were born at close to the expected Mendelian ratios (WT: 19%, n = 20; Alfy GT Het: 52%, n = 54; Alfy GT: 28%, n= 29). Primer pairs spanning the transcript indicated that the full length *Wdfy3* transcript is not produced in Alfy GT mice (*Figure 2—figure supplement 1B*), and antibodies directed against the $NH_3$- or COOH-terminus of Alfy confirmed that full length Alfy protein was not detectable in brain lysates (not shown and *Figure 2B*). This demonstrates that the GT insertion successfully mutagenized the *Wdfy3* locus, leading to the loss of Alfy expression.

The second Alfy-deficient mouse line was produced using a conditional strategy (*Figure 2C*). Mice carrying a conditional *Wdfy3* allele were crossed to $Hprt^{Cre/+}$ females to generate a constitutive null 'Δ' allele (*Figure 2C*, Figure 2—figure supplement 2C) (*Tang et al., 2002*). Similar to Alfy heterozygous the GT insertion, mice heterozygous for the 'Δ' allele (Alfy Het) were healthy and fertile. Breeding of $Wdfy3^{+/\Delta}::Hprt^{Cre/+}$ females with $Wdfy3^{loxP/loxP}::Hprt^{+/Y}$ males generated mice heterozygous (Alfy Het) and homozygous (Alfy KO) for the null allele close to the expected Mendelian ratios (Alfy Het: 56%, n = 40; Alfy KO: 44%, n = 32). Alfy protein also was not detected in Alfy KO mice (*Figure 2D*).

Although neonatal Alfy mutant pups appeared grossly normal, were reactive, and showed no sign of cyanosis, they all died several hours after birth. Unlike their littermates, Alfy mutant pups failed to thrive and consistently showed an absence of milk in their stomachs (*Figure 2E*). Despite being properly cleaned, live Alfy mutant pups were frequently the smallest pups within the litter (*Figure 2F*) and were selectively buried under nesting material, indicating that the dams were rejecting them due to diminished or uncoordinated movement (*Turgeon and Meloche, 2009*). To test this hypothesis, we performed a righting reflex test (*Figure 2G*, *Figure 2—figure supplement 1D*, *Video 1*). Littermates with at least one copy of Alfy were able to right themselves within the allotted test time. In contrast, mice lacking Alfy consistently failed at this test, and demonstrated uncoordinated behavior.

## Alfy is required for the decussation of commissures in the telencephalon

Histological examination of the newborn (P0) brain by Nissl staining revealed that brains from Alfy GT and Alfy KO both demonstrated striking abnormalities in the forebrain, midbrain and hindbrain, including visibly smaller brains and gross enlargement of the lateral ventricles (*Figure 3A,B* and *Figure 3—figure supplement 1A*). Concurrently, there was an apparent loss and disorganization of interhemispheric axonal tracts throughout the brain. Brains of Alfy GT Het or Alfy Het mice were indistinguishable from wildtype brains (*Figure 3A,B*).

To examine the axonal defects further, P0 Alfy mutant brains were stained for neurofilament (NF) (*Figure 3C,D* and *Figure 3—figure supplement 2*). When compared to control, NF labeling of Alfy null brains was greatly reduced, suggesting fewer axonal tracts. A notable abnormality was the failure of the three major forebrain commissures to cross the midline. Axons of the corpus callosum, hippocampal commissure (*Figure 3C*), and anterior commissure (*Figure 3D*) failed to decussate, leading to a separation of the two hemispheres. Two smaller, caudal commissures, the habenular and posterior commissures (*Figure 3—figure supplement 2*), were morphologically abnormal and reduced in size however a portion of their tracts crossed the midline.

We next examined the retinal axon decussation at the optic chiasm by anterograde-labeling with DiI (*Figure 4A–C*). Ipsilateral and contralateral projections were quantified by calculating the ipsilateral index (*Kuwajima et al., 2012*). At P0, the ipsilateral projection in Alfy GT mice (1.40 ± 0.06) was 40% larger than that in wildtype (1.0 ± 0.08) or heterozygous mice (1.1 ± 0.06) (*Figure 4C*). This significant difference suggests that in Alfy mutants, contralateral axons less readily cross the optic chiasm midline. These data reinforce the findings that Alfy is required to properly establish commissures throughout the developing mouse brain.

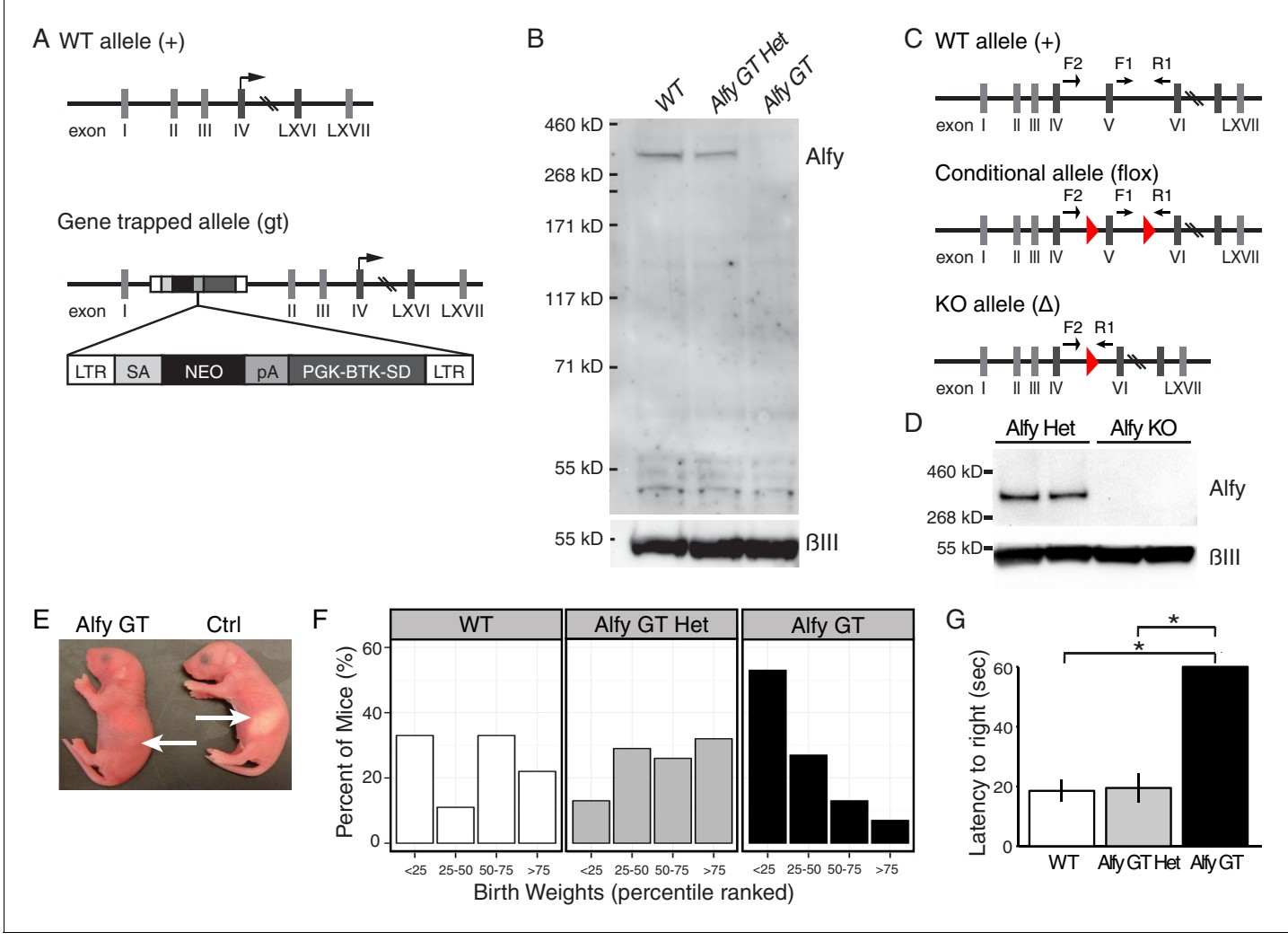

**Figure 2.** Two lines of knockout (KO) mice reveal Alfy is essential for postnatal survival. (**A,B**) Alfy GT mice. (**A**) The GT cassette introduces a splice acceptor site (SA) and causes the premature termination of transcription through the introduction of a poly-adenylation (pA) tail. The translation start site for *Wdfy3* is located in exon IV. RT-PCR indicates that the gene trap insertion leads to a loss of *Wdfy3* transcript (***Figure 2—figure supplement 1A, B***). (**B**) Western blot of brain lysates probed with an antibody against the COOH-terminus of Alfy and βIII (n = 5). Abbreviations: long terminal repeats (LTR), neomycin (NEO), Phosphoglycerate kinase-1 promoter and Bruton tyrosine kinase splice donor site (PGK-BTK-SD). (**C,D**) Alfy KO mice. (**C**) A conditional (flox) *Wdfy3* allele is created by insertion of two loxP sites (red triangles) to flank exon 5, leading to its excision upon exposure to Cre and the creation of the smaller knockout (KO) 'Δ' allele. The two forward and one reverse primers (arrows) used for genotyping are noted (also see ***Figure 2—figure supplement 1C***). (**D**) Immunoblotting detects Alfy in brain lysates from heterozygous (Alfy Het, n = 6) but not in Alfy KO mice (n = 7). (**E–G**) Characterization of newborn Alfy GT mice. (**E**) Newborn Alfy GT and littermatecontrol (Ctrl) mice. Arrows highlight that control pups have stomachs full of milk whereas their GT littermates do not. (**F**) Alfy GT pups are consistently smaller than control. After genotyping, the weights of the animals were percentile ranked and binned into four groups. Approximately 50% of Alfy GT mice (8/15) had birth weights in the lowest 25th percentile, whereas heterozygous and wildtype littermates made up 92% (12/13) of mice with birth weights in the 75th percentile or greater. Data were collected from 10 litters of mice, n = 55. Similar differences were observed between the Alfy KO pups and their heterozygous littermates (data not shown). (**G**) Alfy GT pups lack a righting reflex. The amount of time it took each pup to perform the task was averaged over three trials and recorded as the latency to right. Alfy GT failed to rotate from back to their bellies within 60 sec. A single factor ANOVA revealed genotype had a significant effect on pup behavior (n = 6 WT, 9 Alfy GT Het, 7 Alfy GT; $F_{(2,17)}$ = 20.90, p < 0.001). Fisher's PLSD test indicated a significant difference between Alfy GT and WT (p < 0.001) or heterozygous (p < 0.001) littermates. Bars represent mean ± SEM. Similar differences were observed between Alfy KO pups and their heterozygous littermates (***Figure 2—figure supplement 1D***).

The following figure supplement is available for figure 2:

**Figure supplement 1.** Creation and characterization of Alfy GT and Alfy KO mice.

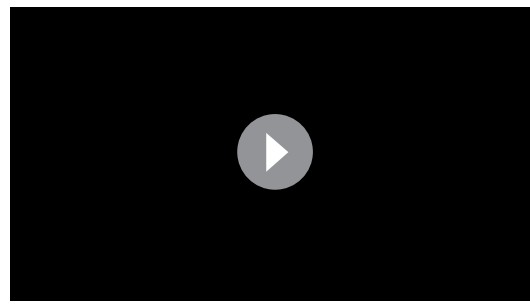

**Video 1.** Mice lacking Alfy have an abnormal righting reflex.

## Glial guidepost cells are mislocalized in *Alfy* KO brains

Two cell populations that help axons navigate the developing forebrain during the genesis of the corpus callosum are the glial wedge (gw) and indusium griseum (ig) glia (*Shu and Richards, 2001*; *Shu et al., 2003*). In light of the presence of Alfy in astroglial cells, as well as the midline crossing defects, we hypothesized that migration abnormalities could contribute to the connectivity defects in Alfy GT and KO mice. Thus, we next examined the callosal guidepost cells, a key population implicated in providing intermediate guidance cues to developing cortical axons. Immunostaining against GFAP on P0 mouse brains revealed abnormalities in Alfy GT brains (*Figure 4D,E*). Unlike control brain, the ig cells were undetectable and the gw appeared disorganized and loosely packed. The midline zipper glia (mz), cells proposed to promote fusion of the two telencephalic hemispheres (*Silver, 1993*), were present but distributed abnormally (*Figure 4D*), whereas cells in the fimbria and dentate gyrus of the hippocampus appear largely normal (*Figure 4E*). Taken together, the disorganization of the ig, gw and mz is consistent with the agenesis of the corpus callosum observed in the absence of Alfy. These findings suggest that cells that act as important guideposts for decussating callosal axons (*Shu and Richards, 2001*) fail to migrate properly in the absence of Alfy, contributing to the connectivity defects observed.

In addition to the localization of the midline glial cells, we also determined if intrinsic alterations in the proliferation or differentiation of cortical projection neurons were also contributing to the disruptions in forebrain axonal connectivity. Nissl staining revealed that cortical layering appeared normal, however there was a significant but modest thinning of the cortical plate/subplate at P0 (*Figure 4—figure supplement 1A*). Although this may be due to the diminished NF staining observed in *Figure 3*, we next determined if cell proliferation was affected. Pulse-labeling with bromo-6-deoxy-uridine (BrdU) at E15.5 revealed cell division in the developing cortex at the expected location in Alfy GT mice (*Figure 4—figure supplement 1B*). To determine if the amount of proliferation was impacted by the loss of Alfy, stereology was performed on BrdU labeled embryos at ages E13.5 and E15.5 (*Figure 4—figure supplement 1C*). No significant difference across genotypes was detected at E13.5, but a modest yet significant reduction in the number of BrdU-positive cells was detected in Alfy KO embryos at E15.5. This difference appeared limited to the cortex, since staining against Ki67 within the subventricular zone at P0 revealed no significant difference between Alfy null mice and controls (*Figure 4—figure supplement 1D*). Further, despite this modest decrease in embryonic proliferation, there were no detectable differences across genotypes in the amount of cell death in E15.5 brains, as indicated by immunohistochemistry against cleaved caspase-3 (*Figure 4—figure supplement 1E*).

We next sought to determine if the loss of Alfy disrupted proliferation in the embryonic cerebral cortex. Although qualitative immunofluorescence against expression markers for immature migrating neurons (DCX) and for early born layer VI neurons (Tbr1) also were not significantly altered in the absence of Alfy, histological staining revealed focal cortical malformations known as focal cortical dysplasias (FCD) present throughout the cerebral cortex of Alfy KO mice, as described previously in a *Wdfy3* hypomorph (*Figure 4—figure supplement 2C–E*) (*Orosco et al., 2014*). These structures were observed as early as E15.5. Thus, the loss of Alfy leads to subtle defects in proliferation, but no changes in cell death or specification in the developing cortical projection neurons. Although columnar organization of discrete regions of the cerebral cortex can be disrupted by the absence of Alfy during embryonic development, it is unlikely that these defects can account for the midline crossing defects observed in Alfy null mice.

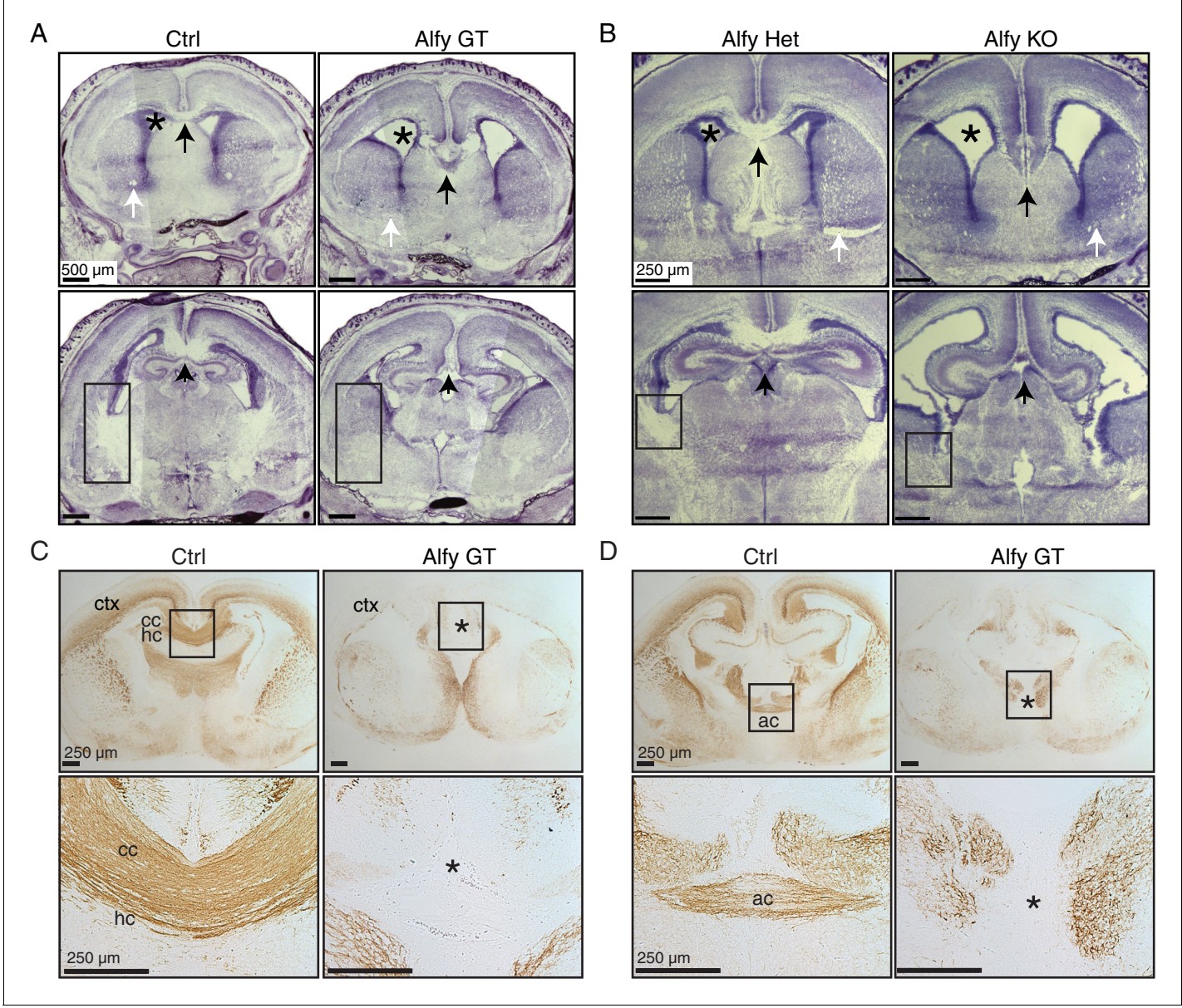

**Figure 3.** Commissures fail to cross the midline appropriately in *Alfy* GT and KO brains. (A,B) Nissl stained forebrain from newborn (P0) control or Alfy mutant (Alfy GT or Alfy KO) mice. Coronal sections are shown. Alfy mutants have many abnormal features, including enlarged ventricles (*), and apparent white matter abnormalities, including absence of a corpus callosum (cc) at the midline (black arrow), undetectable anterior commissure (ac, white arrow), and dysmorphic internal capsule (rectangle). More caudal sections can be found in *Figure 3—figure supplement 1A*. n = 6 WT, 3 Alfy GT Het, 7 Alfy GT; n = 3 Alfy Het, 3 Alfy KO. (C,D) Immunostaining highlights axonal abnormalities in Alfy mutant mice. Coronal sections are shown. The three major forebrain commissures fail to cross the midline in Alfy mutant brains. (C) The cc, hippocampal commissure (hc) and (D) ac are absent in Alfy mutants. '*' denotes midline and lack of axonal connectivity. Boxed regions are shown enlarged below. n = 5/genotype. The habenular and posterior commissures can be found in *Figure 3—figure supplement 2*.

The following figure supplements are available for figure 3:

**Figure supplement 1.** Alfy mutant mice have aberrant and disorganized axon projections.

**Figure supplement 2.** Aberrant and disorganized projections of the habenular and posterior commisural axons.

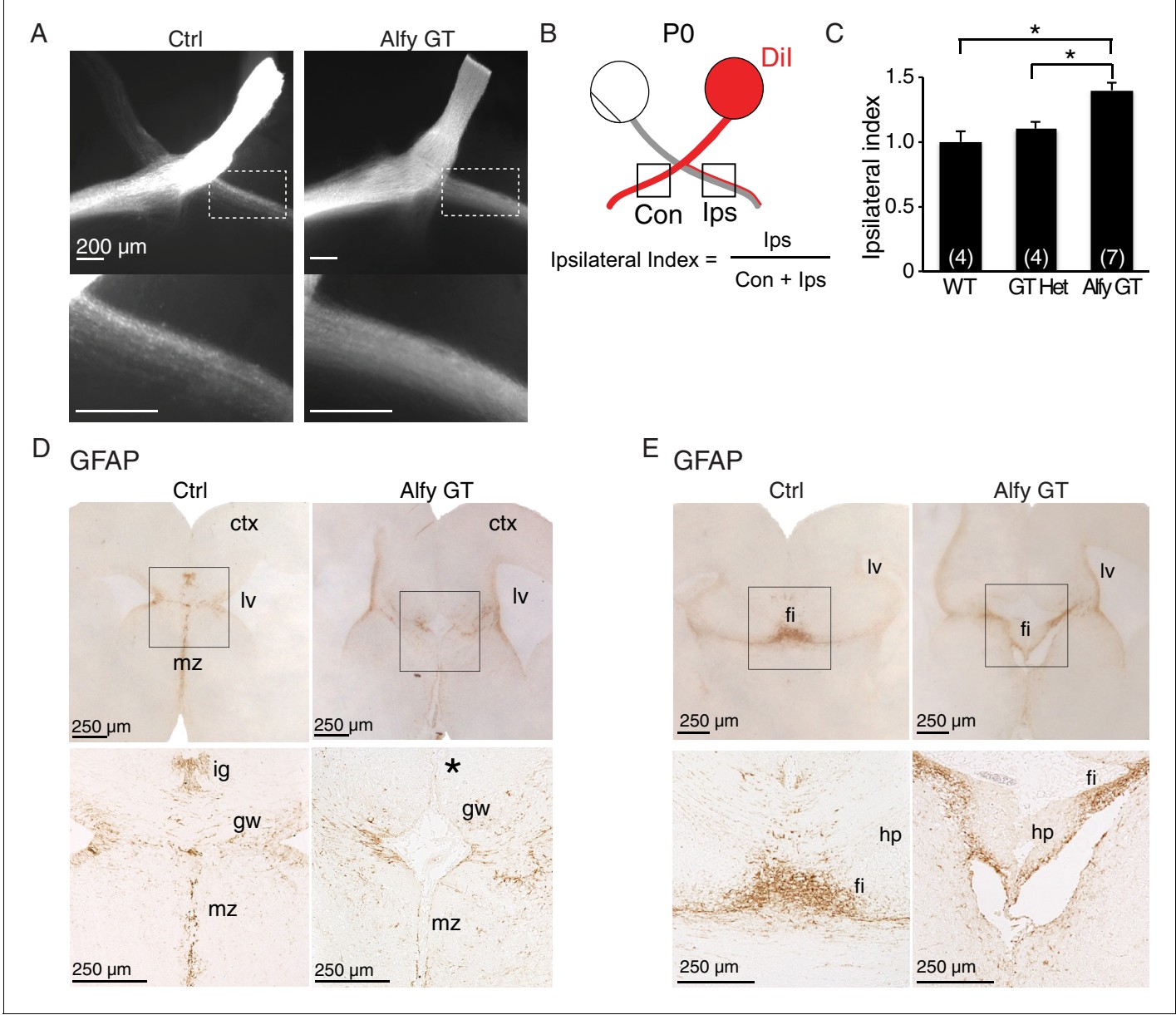

**Figure 4.** Midline crossing defects extend to the optic chiasm in Alfy GT and KO brains, possibly due to disrupted localization of guidepost glial cells. (A–C) Retinal decussation defects are observed in Alfy mutant mice. (A) Whole mounts of P0 optic chiasm are unilaterally labeled with DiI at the optic disc. The ipsilateral projection denoted in the white dashed box, which is shown enlarged below. WT and GT heterozygous littermates were indistinguishable. A heterozygous mouse is shown as a control (Ctrl). (B) Schematic representation of how the ipsilateral index is calculated. Pixel intensities of contralateral and ipsilateral optic tracts are measured within 500 x 500 $\mu m^2$. (C) The ipsilateral index of Alfy mutant mice is greater than both WT and heterozygous (GT Het) littermate controls, indicating the abnormally enlarged ipsilateral projection. Bars represent mean ± SEM, and statistical analysis was determined using one-way ANOVA followed by the Tukey's *post hoc* test. Respective n-values per genotype are noted in the bars. *$p < 0.05$. (D,E) Immunostaining for GFAP in P0 forebrain reveals mislocalization of guidepost glial cells. Coronal sections are shown. (D) The glial wedge (gw), the indusium griseum (ig) and midline zipper (mz) glia populations of glial cells were in the expected locations in WT littermate brains (Ctrl). In Alfy GT brains, the ig was not detectable (expected location denoted by '*'), and the gw and mz cells were disorganized. n = 3/genotype. (E) Glial populations within the fimbria (fi) and dentate gyrus (dg) of the hippocampus were apparent in both genotypes. Boxed areas are shown at higher magnification below. n = 3/genotype. The examination of other potential intrinsic alterations that may contribute to forebrain axonal connectivity can be found in *Figure 4—figure supplements 1* and *2*.

The following figure supplements are available for figure 4:

**Figure supplement 1.** The loss of Alfy leads to modest changes in proliferation but not cell death in the neocortex.

*Figure 4 continued*

**Figure supplement 2.** Focal cortical dysplasias are observed in Alfy GT brains.

## Developmental axonal connectivity defects in Alfy KO brains

In addition to the absence of interhemispheric commissures, another striking feature revealed by NF staining in Alfy GT and KO brains was the evident disorganization and apparent loss of other CNS axon tracts. One particularly prominent defect was found in the organization of the internal capsule, which carries the corticospinal tract, and corticothalamic and thalamocortical projections; the internal capsule of Alfy GT brains was disorganized and followed an unusual trajectory through the striatum (*Figure 3* and *Figure 3—figure supplement 1*).

To gain a temporal perspective on the development of the internal capsule, we examined its development during E15.5 (*Figure 5A*) and E17.5 (*Figure 5B*). NF staining revealed the expected fasciculated axonal projections of the developing internal capsule *en route* to and from the cerebral cortex in controls. In contrast, the thalamocortical projections in Alfy GT mice formed ventrally displaced bundles, with fewer projections observed traveling through the ganglionic eminence. Similarly, Alfy GT embryonic brain sections labeled with the axonal marker NCAML1 also highlighted abnormal bundle-like structures in the ventral diencephalic-telencephalic border (*Figure 5—figure supplement 1*)

We next examined the ventral midbrain (vMB) dopaminergic (DAergic) neurons and their projections to the striatum (*Figure 5C,D*). Tyrosine hydroxylase (TH) staining revealed that the projections and morphology of vMB DAergic neurons were clearly abnormal in the P0 Alfy KO brains. Notably, the A9 and A10 cell populations, which represent the ventral tegmental area (A9) and substantia nigra (A10), revealed that the morphology of the regions had an immature appearance relative to control littermates at the same age (*Figure 5C*). Moreover, rather than follow their typical trajectory through the medium forebrain bundle to striatal targets in the forebrain, in Alfy KO brain, ectopic TH fibers were observed projecting ventrolaterally into the hypothalamus, near the location of the supraoptic decussation (*Figure 5C,D*).

Axonal abnormalities were not limited to the developing brain but were also detected in the developing spinal cord of mice lacking Alfy. At E13.5, disorganized projections into gray matter and the loss of axon bundles were evident in the cervical, thoracic and lumbar levels of the spinal cord in the absence of Alfy, but not in control littermates (*Figure 5E*). In summary, these data indicate that Alfy is essential for establishing axonal tracts throughout the CNS, from forebrain commissures to the spinal cord.

## Attenuated responsiveness to guidance cues in Alfy KO axons

Using two independent mouse models, we find that the loss of Alfy leads to erroneous pathfinding throughout the CNS. Several of these findings are suggestive that the anatomical phenotype observed in Alfy mutant mice may in part be due to defects in axon guidance. For example, an abnormal dopaminergic commissure at the level of the diencephalon traveling through the hypothalamus from the MFB has been previously reported in the *Slit1/Slit2* double KO (*Bagri et al., 2002*), the *Deleted in colorectal cancer (Dcc)* KO (*Xu et al., 2010*), and the *NK2 Homeobox 1 (Nkx2.1)* KO (*Kawano et al., 2003*) mice. Moreover, diminished guidance cue responsiveness could also account for the failure of the midline astroglia to migrate properly. We therefore hypothesized that Alfy may be involved in the ability of neural cells to respond to guidance cues.

Prior to determining if Alfy is involved in axon guidance, we first determined where Alfy is expressed. To do so, we reintroduced full length Alfy into Alfy null primary dissociated cultures to visualize its localization (*Figure 6A*, *Figure 6—figure supplement 1*). Full length Alfy constructs with an N-terminus tag were transfected into DIV7 dissociated primary cortical cultures. Alfy localized throughout the neuron, including within the axon. Neighboring astroglia also showed a diffuse localization of Alfy (*Figure 6—figure supplement 1B*). Next, we performed subcellular fractionation of adult cortical lysates (*Figure 6B,C*, *Figure 6—figure supplement 2*) (*Hallett et al., 2008*). Fractionation revealed that Alfy enriched in the light membrane fraction (P3) along with synaptosomal

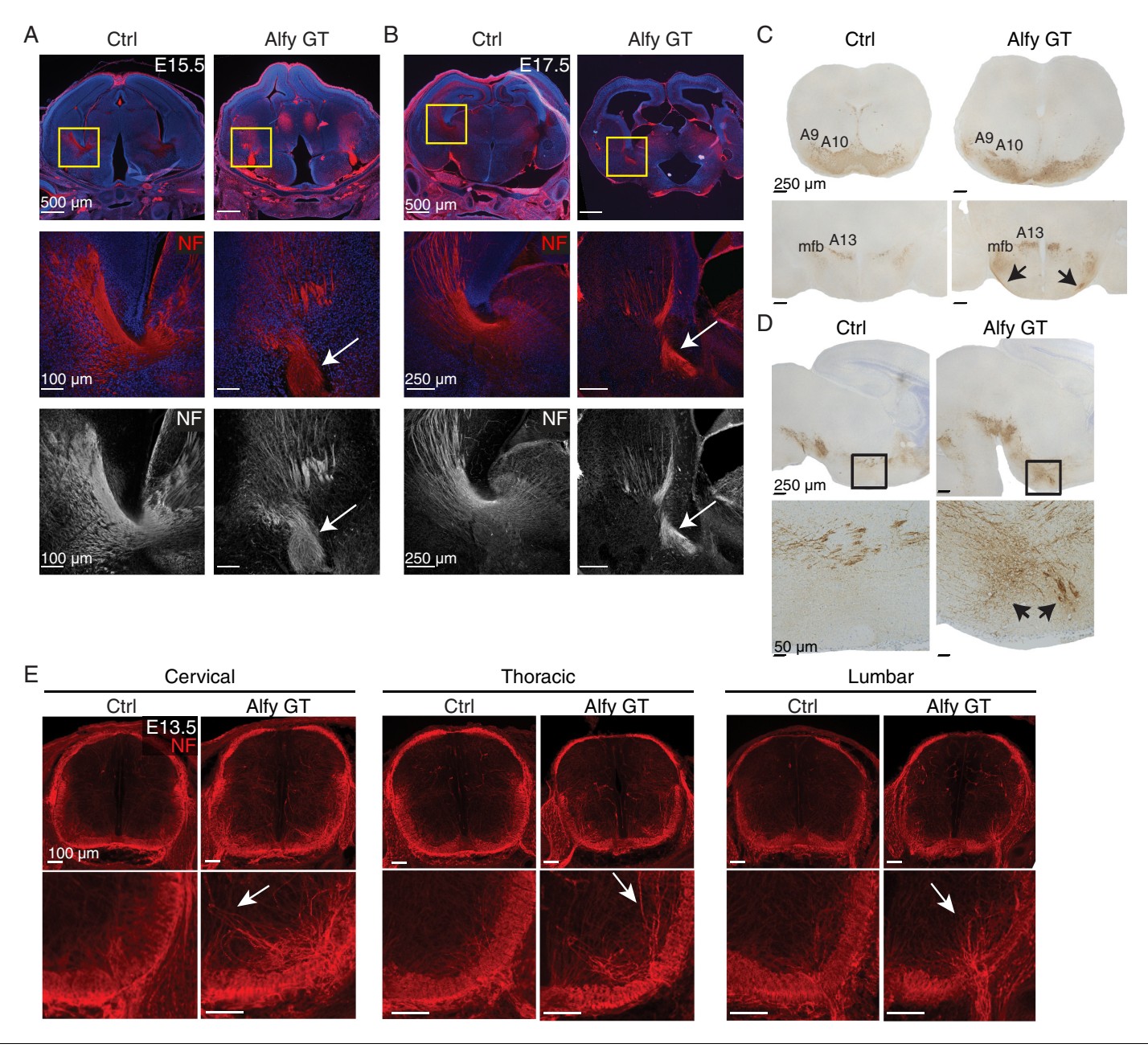

**Figure 5.** Axonal tracts develop abnormally in the Alfy mutant brain and spinal cord. (A,B) Coronal sections of diencephalon at (A) E15.5 (n = 3/genotype) and (B) E17.5 (n = 3/genotype) stained for neurofilament (NF, red) and counter-stained with the nuclear stain Hoechst 33342 (blue). The developing internal capsule is highlighted with a yellow box (top) and higher magnification, confocal images of NF staining are shown with (middle) and without (bottom) Hoechst 33342. In control brains (Ctrl), axons that comprise the internal capsule form a fan-shaped projection of bundled axons within the ganglionic eminence. The abnormal, ventrally displaced, knot-like bundles of axons found in Alfy GT are marked with white arrows. Neural cell adhesion molecular L1 (NCAML1) staining reveals similar defects (*Figure 5—figure supplement 1*). (C,D) TH staining reveals abnormal projections in DAergic cell populations. (C) Coronal sections. A9 and A10 DAergic populations are present in Alfy GT brains appear immature (top). The hypothalamic A13 DAergic cell population and medium forebrain bundle (mfb) are present in the diencephalon in Alfy GT brains; however aberrant projections into the hypothalamus and abnormal midline-crossing events are observed (arrows, bottom). n = 3/genotype. (D) Sagittal sections counterstained with Nissl. In Alfy GT brains, the mfb has ectopic ventral projections into the hypothalamus (arrows, bottom). Boxed regions are shown at higher magnification below. n = 3/genotype. (E) NF staining of E13.5 cervical spinal cord. Coronal sections are shown. Higher magnification images of the dorsal spinal cord are shown below. The abnormal NF patterning observed in Alfy GT embryos is highlighted with white arrows. n = 3/genotype.

The following figure supplement is available for figure 5:

*Figure 5 continued on next page*

*Figure 5 continued*

**Figure supplement 1.** NCAML1 staining confirms axonal defects of the internal capsule in the developing Alfy GT brain.

membrane protein synaptophysin and a classic marker for autophagic vesicles, the membrane bound form of microtubule associated protein light chain 3 (LC3-II) (*Kabeya et al., 2000*) (*Figure 6B,C*). Additionally, Alfy was enriched in the crude synaptic vesicle fraction (LP1). Taken together, these results suggest that Alfy may be involved in regulating the trafficking, sorting or signaling events that are required for brain wiring.

To test this hypothesis, we next explored whether Alfy is required for the outgrowth of axons. Dissociated cultures from P0 cortices were plated and examined for morphological differences at day *in vitro* 7 (DIV7) (*Figure 7A,B*; *Figure 7—figure supplement 1A–C*). Alfy KO neurons displayed healthy growth *in vitro* and Scholl analysis confirmed that they were morphologically similar to control (*Figure 7B*). Staining also revealed that the neuronal cytoskeletal markers Tau-1, βIII Tubulin and microtubule associated protein 2 (MAP2) were expressed comparably across genotypes (data not shown). Taken together, these data indicate Alfy is not required for processes controlling non-directed neuronal outgrowth in a cell autonomous manner.

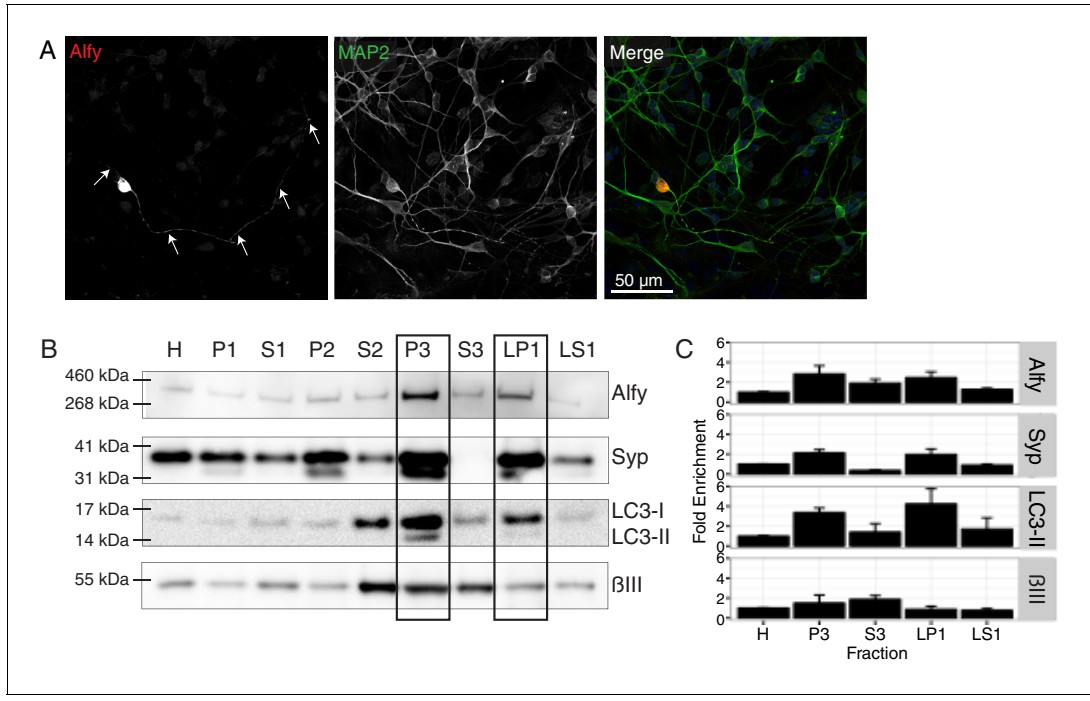

**Figure 6.** Alfy localizes to axons and enriches to membrane fractions. (**A**) Immunofluorescence images showing MAP2-positive neurons expressing Alfy. Merged color image demonstrates co-localization of mCherry-Alfy within a MAP2-positive neuron. Alfy is found within the soma and co-localizes with MAP2 positive projections. Transfections were replicated three times across three independent cultures. Colocalization to βIII Tubulin projections is shown in *Figure 6—figure supplement 1*. (**B,C**) Fractionation of adult cortical lysates reveals that Alfy enriches into membrane fractions. (**B**) Equal amounts of protein per fraction were analyzed by immunoblotting. Alfy was present in membrane fractions that also enriched with LC3-II and synaptophysin (P3, LP1; boxes. (**C**) The fold enrichment measured relative to the total homogenate fraction (H). Bars represent mean enrichment (n = 3) ± SEM. A schematic depiction of the fractionation can be found in *Figure 6—figure supplement 2*.

The following figure supplements are available for figure 6:

**Figure supplement 1.** Alfy is expressed in the soma and axons of neurons.

**Figure supplement 2.** Alfy enriches in membrane fractions from brain.

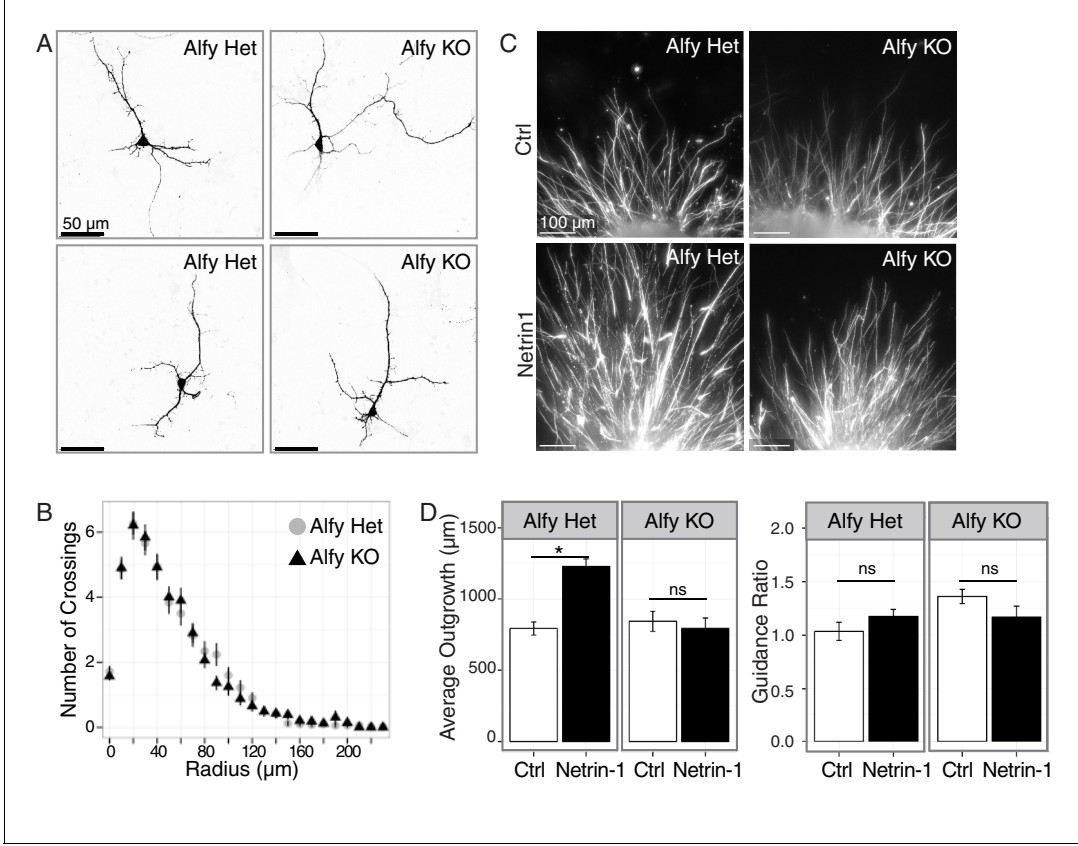

**Figure 7.** Alfy is required for the ability of axons to respond to Netrin-1. (**A,B**) Primary cortical neurons are morphologically similar across genotype. (**A**) Individual, GFP-transfected neurons were imaged by confocal microscopy and traced in ImageJ. Representative maximum projection images of two heterozygous (Het, n=5 brains) and two Alfy KO neurons (n = 5 brains) are shown. Neuronal cytoskeletal markers are expressed comparably across genotype (**Figure 7—figure supplement 1A,B**). (**B**) Scholl analysis of neurons at DIV7 as represented in A. One-way ANOVA revealed no effect of genotype on the number of crossings (n = 50 neurons/genotype; $F_{(1,97)}$ = 0.66, p = 0.41). Analysis of branching (**Figure 7—figure supplement 1C**) similarly showed no effect of genotype. (**C,D**) Alfy KO cortical explants have attenuated responsiveness to Netrin-1. (**C**) Representative images of Alfy Het or Alfy KO cortical explants in the presence of control (Ctrl) or Netrin-1 as described (**Figure 7—figure supplement 2**) and stained with βIII Tubulin. (**D**) Quantification of the average length of outgrowth after 72 hr in culture on data collected from 54 explants (Alfy Het treated with Ctrl: n = 15; Alfy Het treated with Netrin-1: n = 12; Alfy KO treated with Ctrl = 12, Alfy KO treated with Netrin-1: n = 15), from two litters of mice. A two-way ANOVA revealed a significant effect of genotype ($F_{(1,50)}$ = 7.69, p < 0.01) and the presence of Netrin-1 ($F_{(1,50)}$ = 7.71, p < 0.01) on outgrowth, as well as a significant interaction between genotype and Netrin-1 exposure ($F_{(1,50)}$ = 12.29, p < 0.001). Fisher PLSD *post hoc* analysis revealed that exposure to Netrin-1 resulted in a significant increase in outgrowth in Alfy Hets (p < 0.001), but not in Alfy KO (p = 0.56). Under control conditions, there is no difference in the amount of outgrowth between the genotypes (p = 0.56). Netrin-1 attractive guidance was also assessed by determining the Guidance Ratio, which is achieved by measuring the amount of outgrowth from the side of the explant closest to the cue-expressing cell block (proximal) over the amount of growth on the side of the explant furthest from the cell block (distal). Under the conditions of our assay, directional growth in response to Netrin-1 was not observed in the control explants (t(25) = 1.98, p = 0.059) or KO explants (t(22) = 1.77, p = 0.091), suggesting that under the conditions of our assay HEK293T cells secrete Netrin-1 above the threshold for selectively promoting attractive guidance. Similar results were achieved after 48 hr in culture. Bars represent mean ± SEM. *, p < 0.001; ns, not significant.

The following figure supplements are available for figure 7:

**Figure supplement 1.** Primary cortical neurons are similar across genotype.

**Figure supplement 2.** Responsiveness to guidance cues is attenuated in Alfy KO axons.

We next examined if Alfy is essential for neurons to respond appropriately to trophic agents, such as the bi-directional guidance cue Netrin-1 (reviewed in [**Leyva-Diaz and Lopez-Bendito, 2013**]). Explants from control and Alfy KO mice were co-cultured with mock transfected or Netrin-1-Myc transfected HEK293T cells embedded in agarose (**Figure 7—figure supplement 2A**). After

48 hr, the distance of growth outward from cortical explants was measured (*Figure 7C,D*, *Figure 7—figure supplement 2B*). In the presence of Netrin-1, control explants demonstrate significantly increased growth whereas Alfy KO explants did not respond, despite maintaining their ability to extend their processes under non-stimulated conditions (*Figure 7D*). Under the conditions used, directional responsiveness could not be ascertained (*Figure 7D*), likely due to the abundance of Netrin-1 expression due to the use of HEK293T cells (*Shirasaki et al., 1996*). These data indicate that Alfy is required for cortical neurons to respond to the trophic effects of Netrin-1. The loss of Alfy did not affect the total levels of the guidance cue receptor DCC (*Figure 7—figure supplement 2C*). Therefore, our data suggest that Alfy function could involve the regulation of intracellular membrane-based signaling events in response to changes in the extracellular milieu of the developing brain.

## Alfy is an adaptor protein for selective macroautophagy

We and others previously found using stable cell lines that Alfy was an adaptor protein for the degradation of aggregated proteins by selective macroautophagy (*Clausen et al., 2010*; *Filimonenko et al., 2010*; *Lystad et al., 2014*). A key step in response to a guidance cue signaling is the recycling and degradation of large signaling protein complexes. Moreover, upon responding to cues, the resultant neuronal remodeling is also accompanied by the local sequestration and turnover of various cellular components, including cytoskeletal proteins. In light of the degradative capacity of macroautophagy, it is possible that this degradative pathway may be involved, and Alfy acts to sequester these cargoes.

As an adaptor protein, the absence of Alfy should not impact macroautophagic degradation overall. Consistent with previous findings, non-selective macroautophagy remains intact in tissues and cells derived from Alfy null mice (*Filimonenko et al., 2010*) (*Figure 8A–D*). Mice lacking Alfy clearly mounted an autophagic response to starvation as indicated by the increased levels of LC3-II in lysates collected from the perinatal liver (*Figure 8A*), which is consistent with the lack of feeding observed in the pups (*Figure 2E*). The presence of LC3 conversion is consistent with the formation of APs, which we find forms in response to starvation in the presence or absence of Alfy (*Figure 8—figure supplement 1A*). Although neurons can properly mount an autophagic response (*Figure 8D*), no abnormal accumulation of LC3-II was observed in the brain (*Figure 8B*), suggesting that autophagosome maturation also is unaffected. Consistent with this finding, macroautophagic degradation of long lived proteins was also independent of Alfy expression (*Figure 8C*).

Since we confirmed that nonselective macroautophagy was unaffected, we next determine if the loss of Alfy impeded the turnover of its substrates. One naturally occurring Alfy substrate are known as aggresome-like induced structures (ALIS) (*Clausen et al., 2010*). Cytosolic ubiquitinated ALIS were formed in MEFs upon mitotic inhibition (*Figure 8E*). In contrast to control MEFs, these structures accumulated more rapidly in the absence of Alfy, suggesting that in its absence, the turnover of ALIS are impeded. Moreover, unlike in MEFs that lack all forms of macroautophagy (*Figure 8—figure supplement 1B*) (*Kuma et al., 2004*), Alfy GT MEFs do not demonstrate basal protein accumulation, suggesting that Alfy is not required for the turnover of all ubiquitinated substrates. Nonetheless, over-expression of a canonical aggregation-prone polyQ protein, a 17 amino acid fragment of the mutant huntingtin protein with 103 glutamines and an eGFP tag (17aahtt(103Q) eGFP) (*Kazantsev et al., 1999*), accumulated more rapidly by the absence of Alfy (*Figure 8—figure supplement 1C, D*). Thus, in the absence of Alfy, the elimination of ubiquitinated proteins by selective macroautophagy, but not basal macroautophagy, is impaired. Taken together with the rest of our findings, we hypothesize that Alfy-mediated sequestration of cargoes may help selectively eliminate cargoes at or around the membrane, thereby affecting the ability of axons to properly respond to guidance cues.

## Discussion

In this study, our genetic models revealed that Alfy influences how growing axons interact with their environment to form stereotyped axonal connectivity in the central nervous system (CNS). The homozygous loss of *Wdfy3* disrupts the formation of many axonal connections, causing agenesis of all three forebrain commissural tracts and leading to profound changes in brain wiring throughout the CNS, including the optic chiasm, internal capsule, cerebral peduncle, medial forebrain bundle

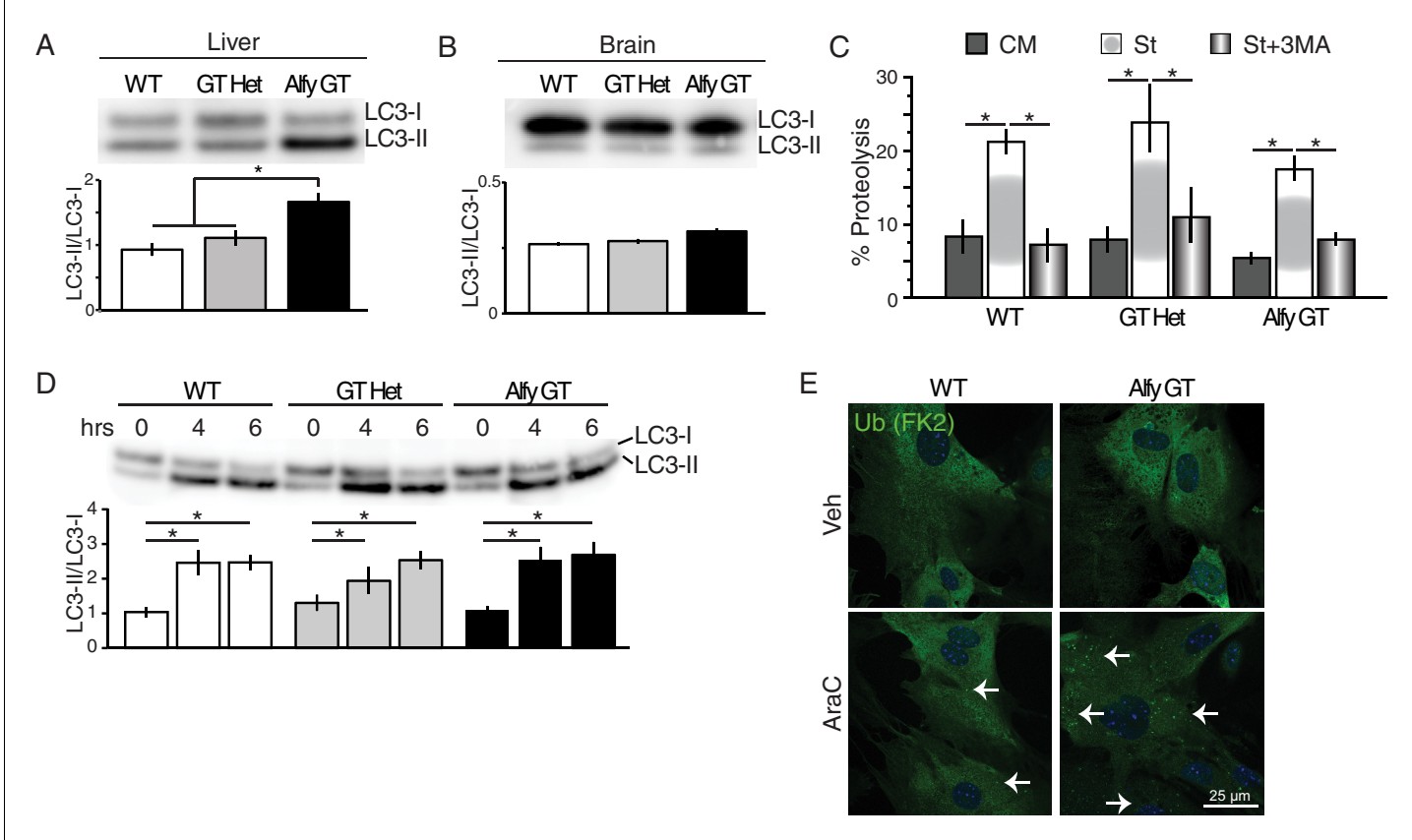

**Figure 8.** Alfy is an adaptor protein for selective macroautophagy (**A–D**) Non-selective macroautophagy proceeds normally in the absence of Alfy. (**A**) Liver lysates from perinatal Alfy GT pups reveal greater LC3 conversion to LC3-II, reflecting the starvation response due to their inability to feed properly (**Figure 2E**). Lysates from normally fed WT and heterozygous littermates mice predominantly contain LC3-I. The bottom graphs quantify LC3 conversion as the ratio of LC3-II/LC3-I. Bars represent mean ± St. Dev. ANOVA reveals a significant effect of genotype on LC3 conversion (n = 7 WT, 6 Alfy GT Het, 6 Alfy GT; $F_{(2,16)}$ = 11.07; $p < 0.001$). Fisher PLSD *post hoc* analyses indicate that LC3 conversion of Alfy GT mice differed significantly from mice with at least one copy of Alfy. (**B**) In the brain, there is no abnormal accumulation of LC3-II in the Alfy GT mice, suggesting that autophagosome maturation proceeds normally. A single-factor ANOVA reveals no significant difference between the three genotypes (n = 5/genotype; $F_{(2,12)}$ = 5.52; $p = 0.90$). (**C**) Macroautophagic protein degradation in response to starvation is normal in the absence of Alfy. Long lived protein degradation assay. (LC3 puncta formation is shown in **Figure 8—figure supplement 1A**). A two-factor ANOVA revealed a significant main effect for treatment (n = 4/genotype; $F_{(2,27)}$ = 24.93, $p < 0.001$), but no significant effect of genotype on percent (%) Proteolysis (n = 4/genotype; $F_{(2,27)}$ = 1.82, $p = 0.18$). Fisher PLSD post hoc analysis revealed % Proteolysis was significantly increased in all genotypes during starvation compared with complete media ($p < 0.001$) and the addition of 5 mM 3MA significantly attenuated the starvation-induced proteolysis ($p < 0.001$). (**D**) In the absence of Alfy, LC3 conversion is normal in primary neurons in response to trophic factor withdrawal (n = 4/genotype). To accumulate LC3-II, cells were starved in the presence of 20 µM leupeptin. (**E**) The loss of Alfy impedes the selective macroautophagic turnover of its cargo, ALIS. Although ubiquitinated structures are readily found when all forms of macroautophagy is impeded (**Figure 8—figure supplement 1B**), the turnover of only specific cargoes is affected by the loss of Alfy. Primary Alfy GT versus Alfy heterozygous (HET) MEFs after 72 hr of Veh or 5 µM AraC. Ubiquitin-positive ALIS bodies form in response to mitotic inhibition, as revealed by FK2 immunofluorescence (green). In the absence of Alfy, these structures accumulate more rapidly (white arrows). n = 3. Similar Alfy dependence can be observed with the aggregation prone expanded polyQ proteins (**Figure 8—figure supplement 1C,D**). For all graphs, bars represent mean ± SEM. ns, not significant; *, $p < 0.05$.

The following figure supplement is available for figure 8:

**Figure supplement 1.** Alfy is an adaptor for selective macroautophagy.

and spinal cord tracts. At the cellular level, the loss of Alfy led to defects in the development of midline glial population, impacted embryonic cortical neuron proliferation and impeded the ability of axons to respond to Netrin-1. Furthermore, Alfy KO mice present a phenotype unlike mice lacking core autophagy genes such as *Atg5*, suggesting the perinatal lethality is not due to a loss of general

autophagic degradation, as Alfy mutant tissue and cells display the classic response to starvation. We hypothesize that Alfy functions as a selective macroautophagy adaptor protein and that Alfy-dependent regulation of intracellular vesicle trafficking and signaling pathways upstream of macro-autophagy can have a significant influence over brain development.

The widespread disruption of axonal wiring identified in the absence of Alfy supports a functional role for this protein in the sequestration and trafficking of substrates into the autophagic pathway. In the adult brain, Alfy facilitates the sorting of ubiquitinated cargo into APs, and one possibility is that during development Alfy has an analogous function. For instance, APs are present in the growth cone and axon tips (*Hollenbeck, 1993*; *Maday and Holzbaur, 2014*; *Maday et al., 2012*). We propose that Alfy could regulate the ability of growth cones to respond to cues by sequestering and sorting proteins into degradative vesicles. Such regulation could be critical for responding to cues in the environment through changes in cell shape and motility. It has been suggested previously that the ubiquitination of guidance receptor complexes could regulate the responsiveness of axons to guidance cues (*O'Donnell et al., 2009*). The trafficking and responsiveness of the axonal guidance receptors Robo1 and UNC5 has been shown to be regulated by the E3 ubiquitin ligases rpm-1 in *C. elegans* and USP33 in murine commissural axons, respectively (*Li et al., 2008*; *Yuasa-Kawada et al., 2009*). Furthermore, deletion of the rpm-1 homologue, Phr1, in mice, causes axonal wiring defects reminiscent of Alfy mutants (*Bloom et al., 2007*). One intriguing possibility is that Alfy might seques-ter cargo ubiquitinated by E3 ligases such as Phr1 or USP33. Alternatively, the association of Alfy with the GABA$_A$ receptor-associated protein (GABARAP) (*Lystad et al., 2014*), implies Alfy could be involved in the selective sorting of plasma membrane receptors. Based on this model, we predict that in the absence of Alfy, the efficient degradation of Alfy-associated cargo would be slowed, or the cargo would be trafficked into alternative pathways, disrupting the temporal regulation of signal-ing events within the growth cone. Similarly, the disruption of the midline glial structures may also reflect the disruption of signaling events that promote the differentiation and migration of these cell populations. Examination of substrates labeled with a Lys63 polyUb chain, which are also associated with APs (*Shaid et al., 2013*), may identify other Alfy-interacting substrates.

*Wdfy3* and its homologue *bchs* have been implicated in trafficking events associated with lyso-somes, yet genetic disruptions yield distinct phenotypes in their respective model systems. Both homologues have been shown to co-localize with markers for degradative vesicles, including APs in mammals and endolysosomes in *Drosophila* (*Lim and Kraut, 2009*; *Lystad et al., 2014*; *Simonsen et al., 2004*) yet, despite the conservation of domain structure, *bchs* loss of function (LoF) mutants have no reported developmental neuroanatomical defects (*Khodosh et al., 2006*). Overex-pression of *bchs,* however can impact synaptic and axonal morphology at the neuromuscular junc-tion and the retina, revealing that *bchs* dosage plays a role in CNS development and function (*Khodosh et al., 2006*; *Kraut et al., 2001*). The difference in phenotype between *bchs* LoF flies and Alfy mutant mice might be explained by the emergence of novel biological functions due to the evo-lutionary expansion of the BEACH family of genes from six in invertebrates to nine in vertebrates (*Cullinane et al., 2013*). Evolutionary divergence also has been described for the BEACH protein, *Neurobeachin (Nbea)*, which has evolved distinct biochemical properties from its *Drosophila* homo-logue *rugose* (*Volders et al., 2012*). We propose Alfy, which shares 48% protein sequence identity with bchs, may have similarly diverged, acquiring new functions that are essential for vertebrate development.

Whereas proteasome-mediated degradation is thought to be the primary catabolic pathway reg-ulating neurodevelopmental signaling (*Yi and Ehlers, 2007*), recent studies indicate that macroau-tophagy may also play a critical role in morphogenesis. For instance, the loss of Ambra1, a regulator of the autophagy protein Beclin1, causes neural tube closure defects in mice (*Fimia et al., 2007*). In humans, recessive mutations in *TECPR2* and *EPG5*, genes implicated in AP maturation (*Stadel et al., 2015*; *Tian et al., 2010*), are responsible, respectively, for spastic paraplegia 49 with brain malforma-tions, including hypogenesis of the corpus callosum [OMIM 615031] and Vici syndrome with brain malformations, including agenesis of the corpus callosum [OMIM 242840] (*Cullup et al., 2013*; *Oz-Levi et al., 2012*). Why loss of these autophagy-related proteins causes corpus callosum defects remains unclear. Although little is known about the importance of core autophagy genes such as *Atg5, Atg7* and *Atg14* in neurodevelopment, no obvious defects in axonal connectivity have been reported (*Hara et al., 2006*; *Komatsu et al., 2006*; *Liang et al., 2010*). One exception is that the conditional elimination of *Atg7* in POMC neurons leads to a defect in postnatal axonal outgrowth

(*Coupe et al., 2012*), and recently, a partial LoF mutation in a core autophagy gene, Atg5, have been reported to lead to cerebellar hypoplasia and developmental delay (*Kim et al., 2016*), but connectivity of the callosum appears unaffected. Given that degradation by autophagy is slower than proteasome-mediated turnover, these data suggest that the sorting and sequestration of cargo, rather than elimination of cargo, is critical for the timing of cellular processes regulated by macroautophagy. Alternatively, as a scaffold for ubiquitinated structures for degradation by macroautophagy, the loss of Alfy might have indirect effects on proteasome activity. For example, Alfy is recruited to aggregated structures that form upon proteasome-inhibition (*Clausen et al., 2010*; *Simonsen et al., 2004*); when Alfy is absent, the persistence of structures that might otherwise be eliminated may impact the efficiency of proteasome-mediated degradation. We have previously reported that the deletion of Alfy does not measurably affect the enzymatic activity of the proteasome (*Filimonenko et al., 2010*) and does not lead to the accumulation of ubiquitinated structures in *Alfy* null MEFs (*Figure 8*). Nonetheless, it is still possible that localized, discrete disruptions of proteasome activity or modifications in activity enough to disrupt timing might be present, leading to the profound phenotype we observe here.

Genetic screening has revealed a possible role for the human *Wdfy3* homolog (*WDFY3*) as a genetic risk factor for intellectual and developmental disabilities. Most recently, a missense mutation in *WDFY3* has been linked to autosomal dominant primary microcephaly (*Kadir et al., 2016*). Furthermore, in autism spectrum disorder (ASD), rare *de novo* nonsense mutations within *WDFY3* have been identified across two independent studies (*De Rubeis et al., 2014*; *Iossifov et al., 2012*). *WDFY3* resides on chromosome 4q21.23 and this region of chromosome 4 was suggested to harbor a genetic predisposition to ASD (*Chen et al., 2006*) and to schizophrenia (*Faraone et al., 2006*; *Paunio et al., 2004*). In addition, the dosage of *WDFY3* is affected in a subset of individuals with a rare chromosome disorder known as 4q21 deletion syndrome [OMIM 613509], characterized by intellectual disabilities (ID), language impairment and congenital birth defects (*Bonnet et al., 2010*; *Cooper et al., 2011*). Taken together, these data imply *WDFY3* is a rare genetic risk factor for childhood neurological disorders. Future studies should aim to determine whether *WDFY3* dosage is critical for human brain development and how mutations could potentially disrupt the biological function of Alfy. A network analysis study on genomic data found that CNVs in genes associated with autophagy were enriched in patients with ASD (*Poultney et al., 2013*). While it remains to be determined whether computational modeling can be used to accurately predict the cellular pathways disrupted in ASD, in light of our findings, we would propose that further study is warranted.

In conclusion, mice lacking Alfy survive embryogenesis, but the loss of Alfy function produces widespread CNS defects in axonal tracts, including loss of the major forebrain commissures. The results of our study provide new insight into the genetic control of brain wiring and open the door to many new questions surrounding the physiological role of Alfy and selective autophagy in CNS development.

## Materials and methods

### Antibodies

The following antibodies were used for western blot, immunocytochemistry, immunohistochemistry, in situ hybridization and immunoprecipitation experiments: Anti-Alfy (rabbit polyclonal anti-COOH Alfy; generously provided by Dr. Anne Simonsen and used for western blots and immunoprecipitation), anti-Alfy-N1 (rabbit polyclonal anti-NH3 Alfy; generously provided by Dr. Masaaki Komatsu and used for western blots), anti-BrdU clone BU-33 (B2531, Sigma-Aldrich [St. Louis, MO]), anti-Caspase 3 active/cleaved form (AB3623, EMD Millipore [Germany]), anti-Class βIII-Tubulin (PRB-435P, Covance [Princeton, NJ]), anti-Human DCC (554222, BD Biosciences [San Jose, CA]), anti-Digoxigenin-AP, Fab fragments (Roche [Switzerland]), anti-Doublecortin (ab18723, Abcam [Cambridge, MA]), anti-GFAP, clone GA5 (MAB3402, EMD Millipore), anti-LC3 (for immunofluorescence, rabbit polyclonal; generously provided by Ron Kopito), anti-LC3B (for western blots, ab48394, Abcam), anti-Ki67 clone B56 (550609, BD Biosciences), anti-MAP-2 (ab11267, Abcam), anti-NCAML1, clone 324 (MAB5272, EMD Millipore), anti-Neurofilament clone2H3 (developed by TM Jessell and J Dodd and was obtained from the Developmental Studies Hybridoma Bank developed under the auspices of the NICHD and maintained by the University of Iowa), anti-Robo1 (H-200, Santa Cruz Biotechnology

[Santa Cruz, CA]), anti-synaptophysin (101 002, Synaptic Systems [Germany]), anti-Tau-1, clone PC1C6 (MAB3420, EMD Millipore), anti-Tbr1 (AB2261, EMD Millipore), and anti-Ubiquitin, clone FK2 (Enzo Life Sciences [Farmingdale, NY]).

## Additional shared reagents

The following reagents were generously provided: Atg5 knockout and wildtype MEFs (Noboru Mizushima, [*Itakura and Mizushima, 2010*]) and MYC-tagged Netrin-1 (pGNET1-myc) (Marc Tessier-Lavigne [*Serafini et al., 1994*]).

## Mice

All animals and procedures complied with the Guide for Care and Use of Laboratory Animals and were approved by the IACUC committee at Columbia University. Mice were maintained in a 12h/12h light/dark cycle in a temperature and humidity controlled environment, with *ad libitum* access to food and water. Alfy GT mice (129/SvEv x C57BL/6; $Wdfy3^{Gt(OSTGST\_5258\_D3)Lex}$) were generated at the Texas Institute for Genomic Medicine (TIGM [College Station, TX]). A complete description of the gene trap vector can be found in (*Zambrowicz et al., 2003*). The Alfy GT mice used in this study were derived from mating heterozygous mice. *Wdfy3* floxed mice were generated on a 129/SvEv x C57BL/6 background under contract with the University of Connecticut. The floxed *Wdfy3* allele has *loxP* sites flanking exon V that is predicted to produce a 66 amino acid peptide following Cre-mediated recombination, instead of the 3508 amino acid full-length Alfy protein. The *Wdfy3* deletion allele (Δ) is created when male mice carrying *Wdfy3* floxed alleles are crossed with Bl6/J female carrier of $Hprt^{Cre/+}$, a Cre-deleter stain with 100% Cre-mediated recombination in oocytes (*Tang et al., 2002*). All experiments included control and experimental littermates and represent data across multiple litters. Genomic DNA was extracted and the alleles were identified using the following primers: Common Alfy forward: 5'-CTTGTTACACTTGTCCCACAGC-3', Alfy WT reverse 5'– TTAGACTTC TAAGCCCACGAGTACC-3', and Alfy GT reverse: 5'-ATAAACCCTCTTGCAGTTGCATC-3'. Genotyping for the Alfy WT, loxP and Δ alleles was performed using following the primers in a multiplex PCR reaction: LoxP-F 5'-gaaagcaagctcgtttacgg-3', Frt-R 5'-aggttaccagccacaaccag -3', and Frt-F 5'-acttgg-gaagagggaagctc-3'.

## Reverse transcription – polymerase chain reaction (RT-PCR)

RNA from whole embryo was isolated using Trizol reagent (Life Technologies [Carlsbad, CA]) according to the manufacture's recommendation. This RNA was used in a reverse transcription reaction containing random 9mers (NEB Biosciences) and reverse transcriptase (Superscript, Qiagen [Germany]) according to manufacturer's instructions. Negative controls (minus reverse transcriptase) were run for each sample. The resulting cDNA was used to determine the abundance of *Wdfy3* mRNA relative to *GFAP, βIII Tubulin* or *Actin*. The following primers were used: βIII Tubulin: Tuj1-1F: 5'-ctacgacatctgcttccgca -3', Tuj1-1R: 5'-gaagggaggtggtgactcca-3'; GFAP: GFAP-F: 5'-gagctcaatgaccgctttgc -3', GFAP-R: 5'-tccttggctcgaagctggt -3'.

## Dissociated primary cortical cultures

Primary dissociated cortical cultures were prepared from postnatal day 0 mouse pups (P0). The cortical lobes were dissected in ice cold DMEM/F12 (Life Technologies) containing 10% heat- inactivated FBS and antibiotics. Cortical tissue were trypsinized for 30 min at 37°C, then triturated in fresh 10% FBS DMEM/F12 through a fire-polished glass pipette and filtered through a 40 micron nylon cell strainer (BD Biosciences). For immunocytochemistry, cells were plated at a density of $1.0 \times 10^5$ cells/well onto coverslips coated with 20 μg/mL polyD-lysine (Sigma Aldrich) and 5 μg/mL mouse laminin (Life Technologies). The medium was changed 2 hr after plating to NB media (Neurobasal media [Life Technologies]) containing 0.5 mM L-glutamine, 1X B27 supplement (Life Technologies), 2% heat inactivated FBS (Life Technologies) and 1X antibiotic (Life Technologies).

## Creation of mouse embryonic fibroblast (MEF)

Embryos were collected from deeply anesthetized pregnant dams at E14.5. Uterine horns were placed into ice cold Hank's buffered saline solution (HBSS). Heart, liver and head were removed for genomic DNA extraction, the remaining tissue was trypsinized in equal volume of HBSS and 0.25%

trypsin for 15 min. at 37°C. Samples were triturated with a fire-polished glass pipette, then diluted into MEFs complete media: DMEM (Life Technologies) containing 10% FBS (Life Technologies), 1X L-glutamine (Life Technologies), 0.1 mM beta-mercaptoethanol (Life Technologies), 1X Pen/strep (Life Technologies). Cells were filtered through a 100 micron sieve and plated. In addition to geno-type confirmation, RNA was extracted from the cultures to measure transcript levels and MEF cultures were confirmed to be mycoplasma free (Takara). To starve the MEF cultures, the complete media was aspirated off the cultures, followed by three PBS washes and cultures were placed in HBSS supplemented with 10 mM HEPES for four hours.

## Synaptosome preparation

The subcellular fractionation of the brain tissue protocol was performed as described (*Hallett et al., 2008*). Briefly, adult control mice were euthanized and cerebral cortex was rapidly dissected and fro-zen on dry ice. An aliquot of each fraction was saved for analysis. Brain tissue was homogenized in TEVP buffer (10 mM Tris, 5 mM NaF, 1 mM $Na_3VO_4$, 1 mM EDTA, 1 mM EGTA, pH. 7.4) containing 320 mM sucrose. Homogenates (H) were fractionated sequentially as shown in *Figure 6—supple-ment figure 2* at 800 *xg*, resulting in the first supernatant (S1) and pellet (P1) fractions, and 9200 *xg* to generate the second supernatant (S2) and crude synaptosomal membranes (P2). P2 was resus-pended in TEVP buffer containing 35.6 mM sucrose for 30 min on ice to release vesicles and organ-elles. P2 was then spun at 25,000 *xg* to segregate the supernatant (LS1) from the membrane fraction (LP1). Lastly, S2 was centrifuged for 2 hr at 165,000 *xg* to generate the supernatant fraction (S3) and light membrane fraction (P3). Protein quantification was performed on all fractions and equal amounts of protein were loaded onto SDS-PAGE.

## Starvation and aggregation experiments

Alfy MEFs as well as Atg5 knockout and wildtype MEFs were maintained in MEF complete media. Starvation was achieved using a starvation medium of HBSS + 10 mM HEPES for 4 hr. To slow lyso-some-mediated degradation, cells were treated with 20 µM leupeptin. Mitotic inhibition was achieved by treating cells with 10 µM arabinofuranosylcytidine (AraC) for 72 hr. Transfections of cells were achieved with Lipofectamine 2000 (Life Technologies) in serum-free media. Cells were trans-fected with constructs encoded by 17aahtt103Q tagged with eGFP as previously described (*Filimonenko et al., 2010*). Cells were fixed 24, 48 and 72 hr later and imaged by epifluorescence microscopy. Long lived protein degradation assay was performed as previously described (*Filimonenko et al., 2010*).

## Western blotting

Whole cell lysates and tissue lysates from freshly dissected brain and liver were generated using a modified RIPA buffer (50 mM Tris-HCl, pH 7.4, 150 mM NaCl, 10 mM EDTA, 0.1% Triton-X 100) con-taining protease and protein phosphatase inhibitors (Halt Inhibitor Cocktail, Roche). Ten strokes in a dounce homogenizer were used to extract protein from tissues and samples were cleared by centri-fugation at 15,000 *xg* for 60 min at 4°C unless otherwise indicated. Protein concentration was deter-mined using the DC Protein Assay (Bio-rad Laboratories [Hercules, CA]) and equal amounts of protein were prepared and loaded onto 4–12% Bis-Tris or 3–8% Tris-Acetate NuPAGE gels and blot-ted to PVDF membranes as described by the manufacture's recommendations (Life Technologies). PVDF membranes were blocked in 5% BSA in TBS containing 1% Tween-20 (TBST). Antibodies were diluted in 1% BSA in TBST and incubated overnight at 4°C. The appropriate HRP-conjugated sec-ondary antibodies (ThermoFisher Scientific) were diluted in 1% BSA TBST and a chemiluminescent reaction (West Dura SuperSignal, ThermoFisher Scientific) was detected by the Versadoc Imaging System (Bio-rad).

## *In situ* hybridization

Fresh frozen tissue was sectioned at 15 µm onto Fisherband Superfrost plus slides and stored at −80°C until use. On day one, slides were warmed at 37°C for 30 min, fixed in 4% paraformaldehyde for 10 min and washed in DECP-treated phosphate buffered saline (PBS). Slides were acetylated with 0.3 M acetic anhydride in 0.1 M triethanolamine for 10 min followed by three PBS washes. Slides were prehybridized in formamide diluted 2X Prehyb buffer (5 M NaCl, 1 M Tris, pH 7.4, 6%

Ficoll, 6% polyvinylpyrrolidone, 6% Bovine serum albumin, 50 mg total Yeast RNA, 5 mg Yeast tRNA, 250 mM EDTA and 50 mg SS DNA) for 1 hr at room temperature. *Wdfy3* riboprobes were prepared from a pCR II plasmid (Life Technologies) containing a PCR fragment of the 5' UTR region of mouse *Wdfy3* from 320–540 nt (ref NM_172882.3). In vitro transcription was performed as described by the manufacturer (Promega Corporation [Madison, WI]) using the T7 polymerase for the antisense probe on template linearized with *KpnI* and Sp6 polymerase for the sense probe on template linearized with *NotI*. Riboprobes were cleaned using the RNeasy MinElute Cleanup Kit (Qiagen) according the manufacturer's protocol. Riboprobes were heated to 80°C for 5 min, quenched on ice and added to 2X Hyb buffer (5 M NaCl, 1 M Tris, 6% Ficoll, 6% polyvinylpyrrolidone, 6% Bovine serum albumin, 50 mg total Yeast RNA, 5 mg Yeast tRNA, 250 mM EDTA, 50 mg SS DNA and 20% dextran sulphate) diluted 1:1 with formamide. Hyb solution was added directly to slides and they were incubated overnight at 68°C in a humidified chamber. On day two, slides were washed in 5X SSC at 68°C for 10 min and coverslips were removed. Three more washes 0.2X SSC were carried out at 68°C, followed by a 0.2X SSC wash at room temperature. Slides were incubated in B1 buffer (0.1M Tris, pH 7.5, 0.15M NaCl) for 5 min then blocked in B1 containing 10% heat-inactivated sheep serum for 1 hr in a humidified chamber. Anti-DIG antibody diluted 1:5000 in B1 buffer containing 1% heat-inactivated sheep serum and slides were incubated overnight at 4°C in a humidified chamber. On day two, slides were washed in B1 buffer three times and equilibrated in B3 buffer (0.1 M Tris, pH 9.5, 0.1 M NaCl, 50 mM MgCl$_2$) for 5 min at room temperature. The staining reaction was carried out using NBT/BCIP (Promega) diluted in B3 buffer in the dark at room temperature overnight in a humidified chamber and the staining reaction was stopped by washing the slides three times in TE (10 mM Tris, pH 8.0; 1 mM EDTA).

## Histology immunohistochemistry
### Early postnatal mice
Deeply anesthetized postnatal mice were perfused with saline and 3.7% paraformaldehyde and post-fixed in paraformaldehyde overnight. Neonatal mouse brains were paraffin embedded and sectioned at 5 μm. H&E and Bielschowsky silver staining was performed by the histology service in the Experimental Molecular Pathology Core facility at Columbia University. Thionin and Hoechst 33,342 were used as counterstains for colorimetric and fluorescent IHC, respectively. The slides were deparaffinized in three xylene washes, followed by sequential ethanol washes to rehydrate, and a 1X Tris buffered saline (TBS) wash. Heat induced antigen retrieval was performed in 10 mM sodium citrate (pH 6.0) for 30 min at 85°C. After allowing the slides to cool, they were washed once in TBS, endogenous peroxidases were blocked in 0.1% hydrogen peroxide for 10 min, followed by three 1 TBS washes. Slides were blocked for 1 hr in 5% BSA TBS containing 0.025% Triton X-100 (TBST). Primary antibodies were diluted in 1% BSA in TBST and incubated overnight at 4°C. The following day slides were washed three times in TBS, followed by incubation in the appropriate secondary antibody (Vector) diluted in 1% BSA in TBST for 1.5 hr at room temperature. For immunofluorescence, after the secondary antibody step, slides were washed three times in TBS, counterstained with Hoechst (Life Technologies) in TBS for 10 min, washed three times in 10 mM Tris, pH 7.4 and coverslipped (Antifade Gold, Life Technologies). For colorimetric reactions, slides were washed three times in TBS, and then incubated with the ABC kits (Vector) for 1 hr at room temperature. The slides were washed in and exposed to a DAB substrate for (6–10 min). After color development, the slides were washed in 10 mM Tris, pH 7.4, serially dehydrated in ethanol, then xylene and then coverslipped.

### Embryonic tissue
Embryonic tissue were collected from deeply anesthetized, timed-pregnant dams. The morning the plug was observed was designated as embryonic stage 0.5. Embryonic brain and spinal cord was dissected and post-fixed for 4 hr, cryoprotected in sucrose overnight and embedded in OCT medium. Free floating cryosections were prepared on a cryostat (Leica CM1950), collected in PBS containing 0.1% sodium azide and stored at 4°C. Immunohistochemistry was carried out essentially as described above, but first the sections were washed three times in TBS, and then continued from the blocking of the endogenous peroxidase step.

## Adult mice

A transcardial perfusion was performed on deeply anesthetized adult mice and tissue was subsequently post-fixed for 4 hr, cryoprotected for 48 hr in sucrose, and embedded in OCT. Free-floating 25 micron cryosections were processed for IHC as described above for an embryonic-free floating tissue.

## Morphology of dissociated primary cortical neurons

On DIV3, cultures were transfected with 1 µg pEGFP-N3 (Clontech) using Lipofectamine 2000 (Life Technologies) in Optimem for 4 hr. The next day, the media was changed to NB media + mitotic inhibitors: 10 µM AraC (Sigma-Aldrich), 10 µM Uridine (Sigma-Aldrich), and 10 µM 5-Fluoro-2'-deoxyuridine (FDU, Sigma-Aldrich) to prevent the excessive growth of glial cells. On DIV7, cells were washed with TBS, fixed for 15 min with 4% paraformaldehyde, permeabilized in 0.1% Triton-X 100 in TBS, blocked in 5% BSA in TBS and stained with primary antibodies diluted in 1% BSA in TBS overnight at 4°C. Alexa Fluor conjugated secondary antibodies (Life Technologies) were diluted in 1% BSA TBS, incubated with coverslips for 1 hr at room temperature, counterstained with Hoechst 33,342 (Life Technologies) and mounted with prolong gold antifade (Life Technologies) onto glass slides. A Leica SP5 confocal microscope was used to image cortical cultures. GFP-transfected cells with the characteristic neuronal morphology, including a small triangular shaped soma were analyzed to compare morphology between Alfy null and Alfy control cultures. Z-stacks were taken every micron using a 20X objective and 3X digital zoom. For each culture, approximately10 individual neurons were imaged totaling 50 control and 49 knockout cortical neurons. Confocal z-stacks were traced and analyzed using the simple neurite tracer segmentation plug-in for the FIJI version of ImageJ (*Longair et al., 2011*).

## Cortical explant cultures and guidance cue responsiveness

Time-pregnant dams were sacrificed in accordance with humane protocols established by the Columbia University IACUC. All dissections were carried out in ice-cold PBS. The embryonic cerebral cortex was dissected out and finely cut into pieces, then 28 gauge needles were used to transfer and position explant pieces inside a drop (5 µL) of Matrigel (BD Biosciences) on a collagen-coated glass bottom petri dish (MatTek). Once explants were placed, 350 µm diameter punches of solidified agarose cell blocks containing either untransfected or Netrin1-MYC transfected HEK293T cells were placed alongside cortical explant pieces. Explants were grown for 48 hr, and then fixed with 4% PFA for 15 min. Explants were imaged and measured using phase contrast microscopy using a 10X objective on a Nikon TiE microscope and NIS Elements software (Nikon). Across two experiments, outgrowth in microns towards the HEK293T cellblock was recorded for the three longest processes in a single plane. The measurements were averaged for each explant and statistical analysis was performed using two-way ANOVA followed by the Fisher LSD post hoc test. Following analysis of outgrowth, explants were further processed for immunocytochemistry as described above. To prepare HEK293T agarose cell blocks, HEK293T cells were seeded at 600,000 cells per six well, then 16 hr later transiently transfected with pGNET1-myc using Lipofectamine 2000. The following day, untransfected HEK293T or HEK293T cells transfected with Netrin1-MYC were trypsinized, pelleted and resuspended in 900 µL of warm (~50°C) 1% agarose dissolved in DMEM. Punches of agarose cell blocks containing HEK293T cells were prepared with a 0.35 mm diameter Harris Uni-Core tissue punch.

## DiI labeling of the optic chiasm

To visualize retinal ganglion cell (RGC) axon projections through the optic chiasm, unilateral whole eye anterograde labeling with DiI (Molecular Probes) was performed on fixed tissue as described previously (*Kuwajima et al., 2012*). Samples were incubated in PBS + 0.02% sodium azide for 14 days at 37°C. Ipsilateral versus contralateral projections were quantified by measuring the pixel intensity of ipsilateral and contralateral optic tracts in a 500 x 500 µm area lateral to the optic chiasm midline with the MetaMorph image analysis software. An ipsilateral index was calculated by dividing the intensity of the ipsilateral projection by the sum of the contralateral and ipsilateral pixel intensities as described previously (*Erskine et al., 2011*; *Kuwajima et al., 2012*). The ipsilateral index in Alfy null mice was normalized to the littermate WT ipsilateral index.

## Proliferation

The number of Ki67 positive cells labeled with DAB within the subventricular zone of the ganglionic eminence was determined by stereology using StereoInvestigator software (MicrobrightField) by an experimenter blind to genotype. For each animal, measurements were made at 60X bilaterally in five matched, adjacent sections spaced 200 μm apart. There were three animals per genotype.

## BrdU injections

Timed pregnant dams were weighed and received an intraperitoneal injection with 50 mg/kg of BrdU at E15.5. One hour following the injection, the dams were deeply anesthetized and the uterine horns were dissected out. Embryo heads were drop fixed in 4% PFA for 48 hr and then cryopro-tected in 30% sucrose, embedded in OCT and sectioned at 25 μm. Sections were processed using an antibody against BrdU as described in the immunohistochemistry section.

## Statistical analyses

Statistical analyses were performed using Statview 5.0 (SAS Institute). Normally distributed data were subject to student t-test, or for multiple comparisons, analysis of variance (ANOVA) followed by the appropriate post hoc comparisons as indicated in the figure legends. Complete F-statistics and calculated p-values are also available in the figure legends. Power analyses for quantitative studies were performed to achieve a power of 0.8 with a confidence of 0.95 using G*Power 3.1 (*Faul et al., 2007*, *2009*). Effect sizes were based upon pilot studies &/or previously published or unpublished materials. n-values are indicative of biological replicates and no data were excluded from analyses.

## Acknowledgements

The authors would like to thank Drs. Wesley Gruber, N Carolyn Schanen and David Sulzer for helpful discussions. Additionally, the authors would like to thank Drs. Vernice Jackson-Lewis, Nikolai Kholo-dilov, and Mikako Sakurai for technical assistance, and the RLPDRD staff for administrative support. This work was supported by NIH RO1 NS077111, RO1 NS063973 and RO1 NS050199 (JB, JD, LMF, MH, MY, AY), The Parkinson's disease foundation (EE, MH, MSY, AY), Hereditary Disease Foundation (LMF) and the Brain & Behavior Research Foundation (JD).

## Additional information

### Funding

| Funder | Grant reference number | Author |
|---|---|---|
| National Institute of Neurological Disorders and Stroke | R01NS077111 | Joanna M Dragich<br>Michael S Yoon<br>Leora M Fox<br>Ai Yamamoto |
| Parkinson's Disease Foundation | | Megumi Hirose-Ikeda<br>Michael S Yoon<br>Evelien Eenjes<br>Ai Yamamoto |
| Hereditary Disease Foundation | | Leora M Fox<br>Ai Yamamoto |
| Brain and Behavior Research Foundation | Young Investigator Grant | Joanna M Dragich |
| National Institute of Neurological Disorders and Stroke | R01NS063973 | Joanna M Dragich<br>Michael S Yoon<br>Joan R Bosco<br>Ai Yamamoto |
| National Institute of Neurological Disorders and Stroke | R01NS050199 | Joanna M Dragich<br>Megumi Hirose-Ikeda<br>Joan R Bosco |

The funders had no role in study design, data collection and interpretation, or the decision to submit the work for publication.

## Author contributions

JMD, AY, Conception and design, Acquisition of data, Analysis and interpretation of data, Drafting or revising the article, Contributed unpublished essential data or reagents; TK, Figure 4A-C, Conception and design, Acquisition of data, Analysis and interpretation of data; MH-I, Conception and design, Acquisition of data, Analysis and interpretation of data, Contributed unpublished essential data or reagents; MSY, Provided essential technical support to Dr. Dragich, the work was done under the direction of Drs. Dragich and Yamamoto, Acquisition of data, Analysis and interpretation of data; EE, LMF, Acquisition of data, Analysis and interpretation of data, Drafting or revising the article; JRB, Provided key technical support via maintenance of the mouse colony, which includes genotyping and setting up the crosses to create the mice necessary for this study, Acquisition of data, Contributed unpublished essential data or reagents; AHL, Provided a contribution of unpublished data which was critical in shaping the direction of this work, and lead to data that was ultimately not added directly to the manuscript, Acquisition of data, Contributed unpublished essential data or reagents; TFO, Acquisition of data, Analysis and interpretation of data, Contributed unpublished essential data or reagents; OY, Conception and design, Acquisition of data, Contributed unpublished essential data or reagents; TM, Characterized one of the Alfy antibodies used in this paper, the antibody was essential for the successful completion of this work and contributed to all of the figures, Contributed unpublished essential data or reagents; SW, Presented unpublished essential data that was critical for directing the in situ hybridization work, as well as providing independent confirmation of some of the findings presented here, Acquisition of data, Contributed unpublished essential data or reagents; YI, Essential characterization of one of the Alfy antibodies used extensively in this study, This is an unpublished reagent, Acquisition of data, Contributed unpublished essential data or reagents; MK, Unpublished findings on an independent Alfy mutant provided invaluable insight into the direction of our studies, and thus indirectly contributed to conception and design, Also the PI directing the creation of one of the Alfy antibodies used in this study, Acquisition of data, Analysis and interpretation of data, Contributed unpublished essential data or reagents; AS, Conception and design, Analysis and interpretation of data, Contributed unpublished essential data or reagents; REB, Analysis and interpretation of data, Drafting or revising the article, Contributed unpublished essential data or reagents; CAM, Conception and design, Analysis and interpretation of data, Drafting or revising the article

## Author ORCIDs

Tomohiro Mita, http://orcid.org/0000-0002-3578-9801
Carol A Mason, http://orcid.org/0000-0001-6253-505X
Ai Yamamoto, http://orcid.org/0000-0002-7059-2449

## Ethics

Animal experimentation: This study was performed in strict accordance with the recommendations in the Guide for the Care and Use of Laboratory Animals of the National Institute of Health. All animals were handled and procedures were approved by the Institute of Animal Care and Use Committee (IACUC) at Columbia University (Protocol number AAAM7902). All care was taken to ensure minimal handling and suffering, as well as to minimize the total number of animals used to successfully complete this study.

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
