## [Decision Letter]

Thank you for submitting your article "The Autophagy linked FYVE protein (Alfy/WDFY3) is Required for Establishing Neuronal Connectivity in the Mammalian Brain" for consideration by *eLife*. Your article has been reviewed by two peer reviewers, including Ralph Nixon, and the evaluation has been overseen by Ivan Dikic as the Reviewing Editor and Senior Editor.

The reviewers have discussed the reviews with one another and the Reviewing Editor has drafted this decision to help you prepare a revised submission.

Summary:

This manuscript reports that Alfy deletion disrupts brain development and especially the guidance of axons and formation of forebrain commissures. Concerns expressed about validating the presence of Alfy in the neuronal and glial populations under investigation have been thoroughly addressed. The evidence presented also supports the conclusion that loss of Alfy disrupts localization of glial guidepost cells and prevents axons from responding to guidance cues such as Netrin-1. Alfy knockout mice display malformations of most major axon tracts in the brain as well as a thinner cerebral cortex and a markedly reduced thalamus. These are novel and worthwhile findings of interest to a broad neuroscience audience and indicate a key developmental role for Alfy. They raise the possibility that the known function of Alfy in selective autophagy may be critical in development. Some support for this latter conclusion is provided by showing that Alfy deletion in the mouse model somewhat impairs selective autophagy of previously established Alfy targets but not macroautophagy as monitored by LC3 behaviour. Overall this manuscript makes an important contribution but there are some important issues that need to be addressed by the authors.

Essential revisions:

1) The authors show that Alfy is expressed in astrocyte cultures. They then look at mixed astrocyte and neuronal cultures and find Alfy expression. They conclude from this that Alfy is expressed in both neurons and glia but this conclusion cannot follow from this experiment. The expression in the mixed culture likely comes from the astrocytes. The data on the mixed culture adds nothing. If the authors want to claim that Alfy is expressed in neurons, which is important for their hypothesis of a cell autonomous role in axonal guidance, then they need to demonstrate expression in neurons that are free of astrocytes.

2) The relationship between the axonal defects and the disruption of selective autophagy of Alfy substrates is correlational and the possibility exists that Alfy serves additional functions that are driving the developmental defects or that deletion elicits a response in another system that is responsible. The authors point out that evidence that the proteasome has been implicated to a greater extent in developmental abnormalities and the possibility of an indirect effect of Alfy deletion on proteasome activity should be discussed. The initial description of Alfy in fact mentioned that proteasome inhibition alters Alfy behavior and it is reasonable to suppose that blocking Alfy function could have some effects on proteasome as compensation. This is one of various possible unconventional roles that might be imagined for a protein with the properties of Alfy. In the absence of direct evidence that selective autophagy block is responsible for the developmental phenotype, it would be advisable to discuss this possibility as an alternative.

3) The authors perform 1 BrdU pulse experiment at E15.5 and find no difference in proliferation by visualisation. They conclude that Alfy knockout mice do not display a defect in proliferation. This data is not comprehensive enough for such a sweeping statement. Analyses should be completed at multiple developmental stages from E12.5 and quantifications performed. The fact that the cortex is thinner and the thalamus is considerably reduced indicates that there is an issue with the generation, or premature differentiation, of neurons in Alfy mutant mice. Although these may not underlie the axonal guidance disorders, the authors cannot present their argument that Alfy specifically regulates axonal guidance and not neuronal development, based on the current evidence provided.

4) In Figure 4—figure supplement 2 the authors nicely demonstrate that Alfy mutant mice display migration defects (although it is not clear what the n value for these observations is). However, the authors conclude from a gross inspection of Doublecortin staining that the mice have no migration defects. This is confusing. Is it because the migration phenotypes occur in only a subset of the mice? It is also not clear what the Tbr1 staining provides? There is no quantification of this result.

5) The axonal guidance assays on explants (Figure 7) show an increase in outgrowth to Netrin1 in control cultures compared with Alfy mutant cultures. This experiment can only conclude an effect on outgrowth, not direction (and therefore not axonal guidance). In order to conclude an effect on axonal guidance, the cue must be derived from a source and a quantification of outgrowth on the side closest to the guidance cue compared to the side away from the guidance cue is conventionally performed.

[Editors' note: further revisions were requested prior to acceptance, as described below.]

Thank you for resubmitting your work entitled "The Autophagy linked FYVE protein (Alfy/WDFY3) is Required for Establishing Neuronal Connectivity in the Mammalian Brain" for further consideration at *eLife*. Your revised article has been favorably evaluated by Ivan Dikic as the Senior editor and two reviewers.

The manuscript has been improved but there are some remaining issues that need to be addressed before acceptance, as outlined below:

The authors need to include the n values in each figure legend for every analysis – even those that are descriptive – so that the reader knows that the result has been replicated.

The new supplementary figures are helpful but please include the plane of section (coronal, sagittal or horizontal) in the figure legend (also for the main figures). For example in Figure 3—figure supplement 2 it’s hard to see how these represent the habenular commissure?

Exact sample numbers have not been included in every figure sub-section (although some have been included). See Figure 2, Figure 4, Figure 7, Figure 8.

---

## [Author Response]

*Essential revisions:*

*1) The authors show that Alfy is expressed in astrocyte cultures. They then look at mixed astrocyte and neuronal cultures and find Alfy expression. They conclude from this that Alfy is expressed in both neurons and glia but this conclusion cannot follow from this experiment. The expression in the mixed culture likely comes from the astrocytes. The data on the mixed culture adds nothing. If the authors want to claim that Alfy is expressed in neurons, which is important for their hypothesis of a cell autonomous role in axonal guidance, then they need to demonstrate expression in neurons that are free of astrocytes.*

In Figure 1—figure supplement 1 we now provide evidence that Alfy is expressed in neurons. To eliminate glial cells, we add mitotic inhibitors to our culture preparation, and maintain the cultures until the glial cells have been eliminated, as measured by the abundance of GFAP by western blot. First, despite the presence of mitotic inhibitors, the amount of BIII tubulin in the cultures remained constant, suggesting that the neuronal population was largely unaffected by this treatment (Figure 1—figure supplement 1). Alfy is present in these predominately neuronal cultures, providing evidence that Alfy is endogenously expressed by neurons. When compared with mixed cultures containing both neurons and glia, we detect lower levels of Alfy, implying that both neuronal and non-neuronal cells produce Alfy. Glial expression of Alfy is supported by additional data included in the Figure 1—figure supplement 2, showing that Alfy is detectable in purified astroglial cultures.

*2) The relationship between the axonal defects and the disruption of selective autophagy of Alfy substrates is correlational and the possibility exists that Alfy serves additional functions that are driving the developmental defects or that deletion elicits a response in another system that is responsible. The authors point out that evidence that the proteasome has been implicated to a greater extent in developmental abnormalities and the possibility of an indirect effect of Alfy deletion on proteasome activity should be discussed. The initial description of Alfy in fact mentioned that proteasome inhibition alters Alfy behavior and it is reasonable to suppose that blocking Alfy function could have some effects on proteasome as compensation. This is one of various possible unconventional roles that might be imagined for a protein with the properties of Alfy. In the absence of direct evidence that selective autophagy block is responsible for the developmental phenotype, it would be advisable to discuss this possibility as an alternative.*

Given that we are uncertain of the molecular mechanism of Alfy leading to this neurodevelopmental phenotype, we agree that a greater discussion of a potential interplay between Alfy expression and proteasome function should be included, especially as an alternative interpretation. This has now been included in the Discussion section, and it reads as follows:

“Alternatively, as a scaffold for ubiquitinated structures for degradation by macroautophagy, the loss of Alfy might have indirect effects on proteasome activity. […] Nonetheless, it is still possible that localized, discrete disruptions of proteasome activity or modifications in activity enough to disrupt timing might be present, leading to the profound phenotype we observe here.”

*3) The authors perform 1 BrdU pulse experiment at E15.5 and find no difference in proliferation by visualisation. They conclude that Alfy knockout mice do not display a defect in proliferation. This data is not comprehensive enough for such a sweeping statement. Analyses should be completed at multiple developmental stages from E12.5 and quantifications performed. The fact that the cortex is thinner and the thalamus is considerably reduced indicate that there is an issue with the generation, or premature differentiation, of neurons in Alfy mutant mice. Although these may not underlie the axonal guidance disorders, the authors cannot present their argument that Alfy specifically regulates axonal guidance and not neuronal development, based on the current evidence provided.*

To address this issue, we now provide stereologic quantification of BrdU incorporated nuclei from E13.5 and E15.5. As now shown in Figure 4—figure supplement 1, we report that the number of proliferating cells in the neocortical ventricular zone labeled by BrdU at E13.5 was similar between Alfy KO and control embryos, but a modest but significant reduction was observed in Alfy KO embryos at E15.5. In light of this addition, we have now updated our Discussion section to reflect this new finding.

*4) In Figure 4—figure supplement 2 the authors nicely demonstrate that Alfy mutant mice display migration defects (although it is not clear what the n value for these observations is). However, the authors conclude from a gross inspection of Doublecortin staining that the mice have no migration defects. This is confusing. Is it because the migration phenotypes occur in only a subset of the mice? It is also not clear what the Tbr1 staining provides? There is no quantification of this result.*

We apologize for the confusion. Our inclusion of the Doublecortin staining and the Tbr1 patterning was to indicate qualitatively that markers indicative of migration were intact, but, as the reviewers indicate, discrete focal cortical dysplasias (FCD) are observed. We clarify this statement accordingly. In addition, we now provide in Figure 4—figure supplement 2 a quantification of the number of animals in which we observe FCDs (which are in all Alfy null animals examined). We also provide new data, indicating that these malformations are already present in E15.5 cortices.

*5) The axonal guidance assays on explants (Figure 7) show an increase in outgrowth to Netrin1 in control cultures compared with Alfy mutant cultures. This experiment can only conclude an effect on outgrowth, not direction (and therefore not axonal guidance). In order to conclude an effect on axonal guidance, the cue must be derived from a source and a quantification of outgrowth on the side closest to the guidance cue compared to the side away from the guidance cue is conventionally performed.*

We previously performed a co-culture assay to investigate whether Alfy regulates axonal outgrowth and guidance in response to extracellular cues in the environment but without the quantification of directional axonal outgrowth from the explant, we cannot conclude that Alfy is involved in regulating axon guidance. In this submission, we include an analysis of directional outgrowth that was calculated by measuring the length of axons growing on the side of the explant facing the cue and dividing by the length of axons on the opposite side of the explant. We represent this as a guidance ratio and find that while our co-culture conditions generate a reproducible effect on outgrowth in response to Netrin-1 exposure, we do not observe directional growth in either genotype, indicating that our culture conditions do not permit this measurement. We now include both measurements in Figure 7, and additional images in Figure 7—figure supplement 2. One explanation for this outcome is that HEK293T cells have a high secretory capacity, which may cause Netrin-1 act more diffusely. In light of these findings, we have updated our Results and Discussion sections to indicate that the Alfy KO neurons respond less robustly to the trophic effects of Netrin-1, but that we cannot conclude that the loss of Alfy causes a defect in axon guidance.

[Editors' note: further revisions were requested prior to acceptance, as described below.]

*The manuscript has been improved but there are some remaining issues that need to be addressed before acceptance, as outlined below:*

*The authors need to include the n values in each figure legend for every analysis – even those that are descriptive – so that the reader knows that the result has been replicated.*

Please find enclosed all of the n values throughout the figure legends, including all subsections. Based on the instructions we have also placed the individual n values (per genotype, for example) in addition to the F‐statistic that has been provided.

*The new supplementary figures are helpful but please include the plane of section (coronal, sagittal or horizontal) in the figure legend (also for the main figures). For example in Figure 3—figure supplement 2 it’s hard to see how these represent the habenular commissure?*

*Exact sample numbers have not been included in every figure sub-section (although some have been included). See Figure 2, Figure 4, Figure 7, Figure 8.*

We have added the plane of sections used in the figures in the figure legends when applicable. There appears to have been some confusion with the habenular commissure (Figure 3—figure supplement 2). We would like to apologise, as we realized at this time that the control was presented upside down relative to the other coronal images that were provided, which may have contributed to the confusion. We have corrected this, and to ensure clarity, we have also provided a lower powered magnification of the section in Figure 3—figure supplement 2.